# Combining Planting Patterns with Mulching Bolsters the Soil Water Content, Growth, Yield, and Water Use Efficiency of Spring Wheat under Limited Water Supply in Arid Regions

Salah El-Hendawy [1,2,*], Bazel Alsamin [1], Nabil Mohammed [1], Nasser Al-Suhaibani [1], Yahya Refay [1], Majed Alotaibi [1], ElKamil Tola [3] and Mohamed A. Mattar [4,5]

1. Department of Plant Production, College of Food and Agriculture Sciences, King Saud University, Riyadh 11451, Saudi Arabia; balsmain@ksu.edu.sa (B.A.); nmohmmed@ksu.edu.sa (N.M.); nsuhaib@ksu.edu.sa (N.A.-S.); refay@ksu.edu.sa (Y.R.); malotaibia@ksu.edu.sa (M.A.)
2. Department of Agronomy, Faculty of Agriculture, Suez Canal University, Ismailia 41522, Egypt
3. Precision Agriculture Research Chair (PARC), Department of Agricultural Engineering, College of Food and Agriculture Sciences, King Saud University, Riyadh 11451, Saudi Arabia; etola@ksu.edu.sa
4. Department of Agricultural Engineering, College of Food and Agriculture Sciences, King Saud University, Riyadh 11451, Saudi Arabia; mmattar@ksu.edu.sa
5. Agricultural Engineering Research Institute (AEnRI), Agricultural Research Centre, Giza 12618, Egypt
* Correspondence: mosalah@ksu.edu.sa; Tel.: +966-535318364

**Abstract:** Innovations in water-saving cultivation strategies are urgently needed to achieve high yield and elevated water use efficiency (WUE) simultaneously in arid regions with limited water resources. Here, we conducted a two-year field study to compare the impacts of eight combinations of planting patterns (PPs) and mulching on the soil water content (SWC) in the top 60 cm soil layer, the growth, the yield, and the WUE of wheat under two irrigation rates (1.00 and 0.50 ET). These combinations included three conventional flat planting (CF) patterns, including CF without mulch (CFNM), with plastic film (CFPM), and with wheat straw mulch (CFSM); three raised-bed planting (RB) patterns, including RB without mulch (RBNM), with plastic film (RBPM), and wheat straw (RBSM) mulch; and two ridge–furrow planting (RF) patterns, including RF without mulch (RFNM) and with plastic film mulch (RFPM). The results showed that the tested treatments affected the SWC at different depths under both irrigation rates. Compared with the two non-mulched treatments under 0.50 ET, the SWC of the three PPs with plastic film and the two PPs with wheat straw mulching were significantly higher before irrigation by 14.4–22.0% and 6.9–17.2% at 0–20 cm soil depth, 16.4–29.0% and 6.6–14.9% at 20–40 cm soil depth, and 3.3–34.8% and 3.4–14.5% at 40–60 cm soil depth, respectively. All measured wheat parameters, except harvest index, were significantly affected by the interaction between irrigation rate and PPs. The highest values for plant dry weight (PDW), yield components, grain yield (GY), and WUE under 1.00 ET were obtained in the two PPs with wheat straw mulch, while the three PPs with plastic film showed the highest values of these parameters under 0.50 ET. The yield response factor (Ky) based on PDW was acceptable for all PPs mulched with plastic film and wheat straw as well as for RFNM, while Ky based on GY was acceptable only for the PPs mulched with plastic film and for RFNM, as the Ky values of these PPs were less than 1 under 0.50 ET. The SWC at different depths exhibited quadratic and nonsignificant relationships with all parameters under 1.00 ET, while these relationships were linear and strong under 0.50 ET, with a few exceptions. Overall, we conclude that combining any PPs with plastic film mulching could be used as a feasible and effective strategy for obtaining high wheat yield and WUE in the irrigated and arid agroecosystem.

**Keywords:** plastic film mulching; production functions; ridge–furrow; wheat straw mulching; water-saving; yield response factor

## 1. Introduction

Water shortage is considered to be the most critical factor that limits the development of sustainable irrigated agriculture in arid and semiarid regions. Water shortage, combined with abrupt climatic changes and continuous population growth, is expected to challenge the sustainability of crop production and the future of food security in these regions. Furthermore, the agriculture sector in these regions consumes approximately 70–75% of the total available freshwater resources, because the cultivation of crops depends solely upon irrigation. Therefore, water shortage events have gained special importance in both political and scientific realms in these regions. Consequently, governments in these regions have issued several regulations to reduce the amount of water allocated to the agriculture sector while maintaining current crop productivity levels through maximizing water productivity (WP). WP can be maximized by shifting the focus from maximizing grain yield (GY) per unit area towards maximizing GY per unit of water applied, as well as by replacing the paradigm of full irrigation rate with limited irrigation rate [1–3]. Therefore, finding ways to significantly reduce the use of irrigation water and increase grain yield simultaneously under arid and semiarid conditions is crucial.

The range of issues mentioned above can be achieved by employing several strategies, including improving the drought tolerance of genotypes to adapt to water deficit stress [4,5]; optimizing the combination of irrigation rate and scheduling to decrease the amount of water that infiltrates the soil beneath the root zone [6–8]; using antitranspirant substances to reduce transpiration, which is responsible for the loss of 95% of the water absorbed by the plant [9,10]; adopting site-specific agricultural practices, such as modifying planting patterns and applying conservation tillage in combination with mulching with different materials, to decrease the amount of water lost from the soil surface through evaporation, which accounts for up to 40% of the total crop water use under arid conditions; increasing soil water storage; extending the period of soil water availability to plants between two consecutive irrigation events; increasing rainfall harvesting and percolation; suppressing weed growth; modifying plant microclimate conditions; and increasing nutrient use efficiency [11–21].

Recently, the combination of planting patterns (PPs) and mulching with different materials, such as plastic film or plant straw, is considered as one of the most important strategies for adopting site-specific agricultural practices. These strategies have gained considerable attention in many countries around the world, because of their vital regulatory role in improving the soil water content (SWC) and the amount of water available to plants, thus improving the GY and water use efficiency (WUE) substantially when compared with non-mulching conventional flat planting. For instance, Zhang et al. [22] reported that the combination of ridge planting and mulching with plastic film (RFPFM) increased the GY and WUE of dry land wheat in the Loess Plateau of China by 30% and 35%, respectively, compared with non-mulched conventional flat planting. Liu et al. [21] also found that the RFPFM planting method increased the GY of wheat by 51.7%, 64.8%, 25.5%, and 5.84% at 0, 400, 1200, and 2000 $m^3$ $ha^{-1}$ irrigation rates, respectively, compared with traditional flatbed planting. In another study, the RFPFM planting method resulted in approximately 74% and 46.3% of the GY and WUE of winter wheat, respectively, under rainfall conditions compared with those obtained under well-irrigated conditions [23]. The RFPFM planting method plays a vital role in improving GY under a wide range of environmental conditions; this PP increased the crop yield by 10–40%, 30–90%, and 50–100% under wet, average rainfall, and drought conditions, respectively, compared with conventional flat planting under normal rainfall conditions [24].

The effectiveness of mulching practices and their differential effects on the GY and WUE of crops also depend on the materials used for covering the soil surface. Straw mulch is widely used not only to conserve the SWC and improve water availability to plants but also to improve soil fertility and soil organic matter content [25,26]. Many studies have shown that wheat straw mulching improved the SWC (0–2 m soil layer), GY, and WUE of wheat by 1–23%, 13–23%, and 24–33%, respectively, compared with

no mulching treatment [27,28]. Zhao et al. [19] found that the SWC and GY of wheat were 17.7–75.9% and 3.2–8% greater with straw mulching, respectively, than without. Noor et al. [29] also reported that the wheat straw mulch (5000 kg ha$^{-1}$) decreased the daytime temperature by 1.9 °C and 1.5 °C, while increased the SWC by 2.5 and 3.0% at 0–15 cm and 15–30 cm soil layers, respectively, compared with no mulching treatment. The low-density polyethylene plastic film is also a common material used as mulches with different field crops in several countries. This material helped to attain high GY and WUE by conserving soil water when used as mulching individually or combined with different PPs. For instance, Mak-Mensah et al. [30] have shown that, compared with no mulching, the plastic film mulching increased GY and WUE of wheat, potato, and maize by an average of 12–75.7% and 12.8–68.5%, respectively, due to the reduction in ET by 0.5–12.8%. In relation to bare ground, plastic mulch increased GY and WUE of wheat by 19% and 26%, respectively [31]. In terms of partial and full plastic film mulching of soil, the increase in GY under the former and the latter treatments was 38.9 and 77.9%, respectively, and the maximum increase in GY (84.7%) was obtained with the combination of full mulching and ridges planting [11]. Comparison of GY and WUE of maize between plastic film and straw mulching, Gao et al. [32] found that the plastic mulching was the best way for enhancing GY (59.6%) and WUE (29.9%) in the fallow period, while the straw mulching was the best way to increase GY (12.4%) and WUE (13.2%) in the growth period. They also reported that plastic film and ridge row mulching was the best way to enhance growth of maize and increase their GY and WUE in the arid region and low temperature area of China. Therefore, the combination of a specific PP with wheat straw and plastic film mulching may help to simultaneously maximize the GY and WUE at low irrigation rates. Additionally, the selection of appropriate mulching materials for maximizing GY and WUE under irrigation conditions in arid regions such as Saudi Arabia is very urgent. Furthermore, the effect of the combination of PPs and mulching with any mulching materials on growth, GY, and WUE has been highly dependent on crop type, and climate and soil conditions [11,13,21,29–33]. As of yet, most studies involving a combination of PP with mulching practices have focused mainly on the rainfed farming system, but few studies have evaluated this combination practices on GY and WP under the irrigated farming systems in arid conditions. This is the first time that such a study has been conducted in Saudi Arabia, a typical arid country.

The relationship between GY and seasonal crop evapotranspiration (ET), which is also known as crop water-yield production function (CWPF), is a practical way to assess the efficiency of irrigation deficit strategies for a given crop under given site-specific agricultural practices [3,6,34]. A linear relationship between GY and ET, especially under irrigation-deficit conditions, indicates that all of the applied irrigation water is used for crop production, whereas a curvilinear relationship indicates that all of the applied irrigation water was not used for crop production as some of it was lost by ET from the surface soil or by deep percolation below the effective rooting zone [6,35,36]. The yield response factor (Ky), which establishes the relationship between ET deficit and GY depression, is another effective tool to examine the response of a given crop to soil water management [3,6,37]. Therefore, both the CWPF and Ky could be used as efficient indicators to judge the potential benefits of the combination of a particular PP and mulching method on the GY and WUE of a crop under different irrigation treatments.

Although the combination of PP with mulching has gained popularity for cultivating food crops in rainfed regions to increase rainwater harvesting and storage in the root zone, this strategy has not yet been promoted widely in irrigated regions. Moreover, information available on the effects of this system on GY and WUE of irrigated spring wheat is not consistent. Therefore, the main objectives of this study were to (1) compare the impacts of different combinations of PPs (flat, raised-bed, and ridge–furrow planting) and mulching (plastic film, wheat straw, and no mulching) on the SWC, growth, yield components, GY, and WUE of spring wheat under irrigated farming systems in arid conditions and (2) identify the most appropriate combination between PPs and mulching with plastic film and wheat straw to achieve maximum GY and WUE simultaneously for irrigated spring

wheat under arid conditions through Ky, CWPF, and WUE–yield relationships. The results of this study are expected to provide novel insight into the appropriate combinations of PPs and mulching that can be used as an effective way to significantly increase GY and decrease the amount of irrigation water needed for agricultural production in the arid agroecosystem with limited water resources.

## 2. Materials and Methods

### 2.1. Site Description

This study was conducted at the Dierab Experimental Station (24°25′ N, 46°34′ E, 600 m above the sea level) of the College of Food and Agriculture Sciences, King Saud University, Saudi Arabia over two spring wheat growing seasons, 2019–2020 (season 1) and 2020–2021 (season 2). The experimental site has a typical arid climate with limited rainfall (approximately 30 mm annually). The average of maximum/minimum temperature, precipitation, and humidity were measured every 15 days between December and April of the two spring wheat-growing seasons (Figure 1). Values of average maximum/minimum temperature were similar in both seasons, while those of humidity and precipitation showed slight variation between the two seasons, especially in the first half of January and second halves of March and April (Figure 1). The top 1.0 m of soil is sandy loam (56.70% sand, 28.40% silt, and 14.90% clay), with the following physical and chemical properties: mean bulk density, 1.48 g cm$^{-3}$; average organic matter, 0.46%; average field water-holding capacity, 18.89%; average permanent wilting point, 7.28%; available N, 45.2 mg kg$^{-1}$; available P, 7.44 mg kg$^{-1}$; and available K, 186.9 mg kg$^{-1}$ [38].

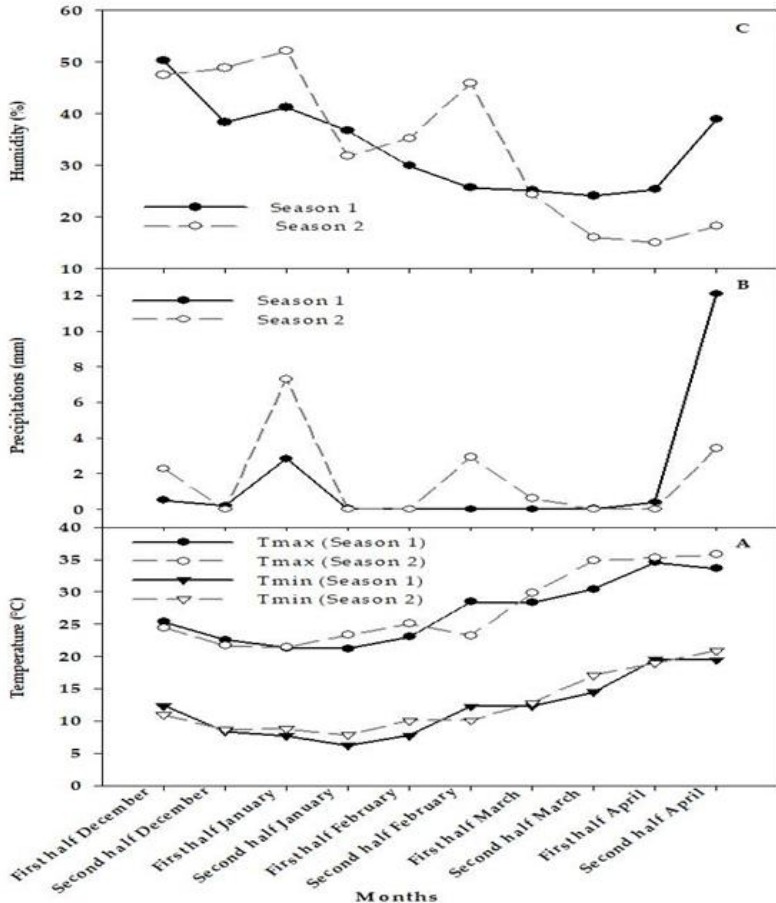

**Figure 1.** Average of maximum (Tmax) minimum (Tmin) temperature (**A**), precipitation (**B**), and humidity (**C**) during 2019–2020 (season 1) and 2020–2021 (season 2) of spring wheat-growing seasons at Dierab Experimental Station.

## 2.2. Experimental Design and Field Management

The different treatments were laid out in a randomized complete block design with split-plot arrangement and three replications. Two irrigation rates were applied to the main plots: full (1.00 ET) and limited (0.50 ET). Eight combinations of the PP and mulching were applied to subplots: (1) conventional flat planting (CF) without mulching (CFNM); (2) CF with plastic film mulch (CFPM); (3) CF with wheat straw mulch (CFSM); (4) raised-bed planting (RB) without mulch (RBNM); (5) RB with plastic film mulch (RBPM); (6) RB with wheat straw mulch (RBSM); (7) ridge–furrow planting (RF) without mulch (RFNM); and (8) RF with plastic film mulch on both ridges and furrows (RFPM) (Figure 2).

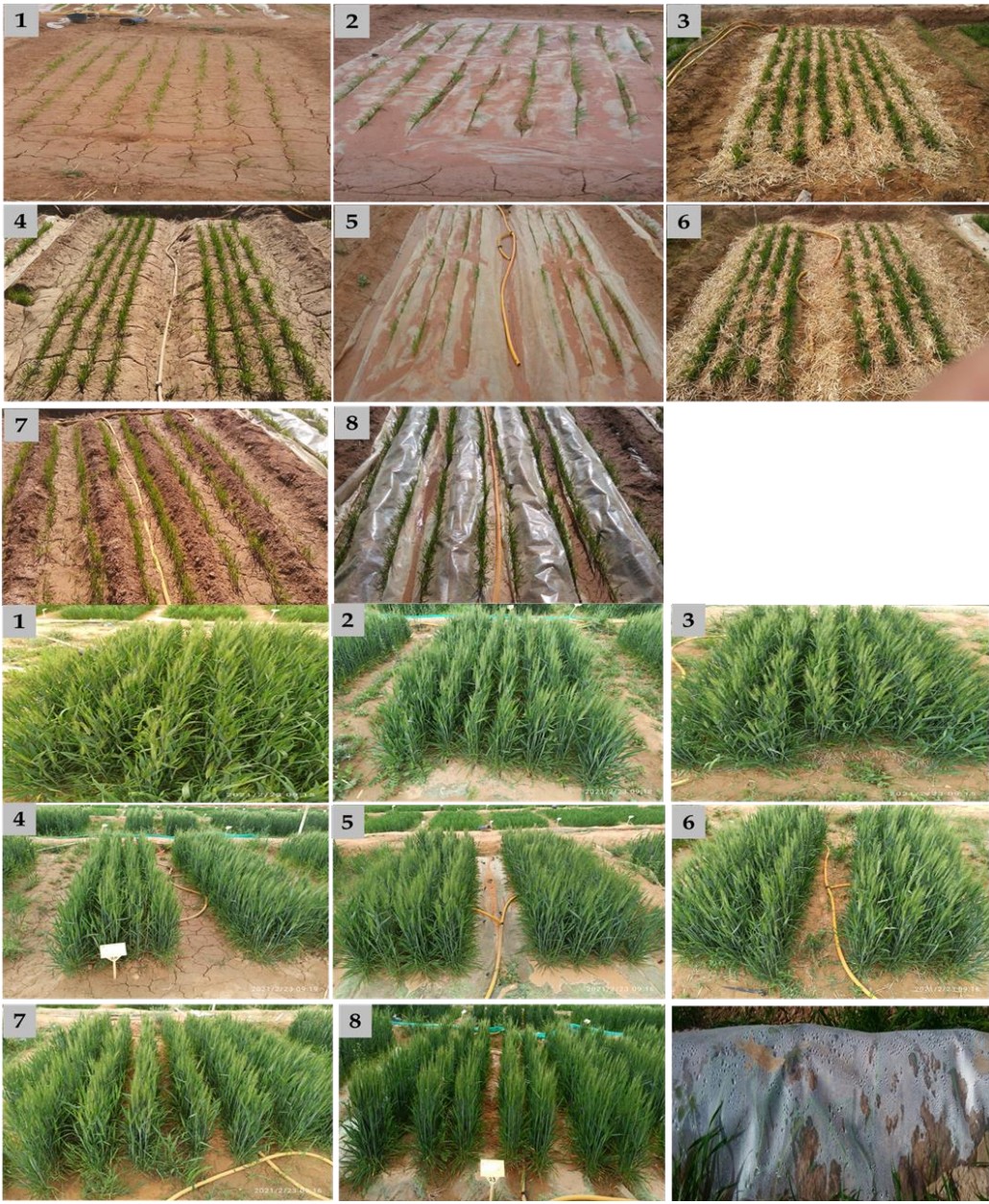

**Figure 2.** Plot layouts of different combination of planting patterns and mulching at seedling and heading growth stages. (**1**) Conventional flat planting (CF) without mulching (CFNM); (**2**) CF with plastic film mulch (CFPM); (**3**) CF with wheat straw mulch (CFSM); (**4**) raised-bed planting (RB) without mulch (RBNM); (**5**) RB with plastic film mulch (RBPM); (**6**) RB with wheat straw mulch (RBSM); (**7**) ridge–furrow planting (RF) without mulch (RFNM); and (**8**) RF with plastic film mulch on both ridges and furrows (RFPM).

Two weeks before planting, the field was prepared by plowing the soil three times to a depth of 25 cm, followed by disk harrowing once to a depth of 10 cm. Then, the soil was leveled, and the field was divided into two main plots (70 m × 4 m each), with a 3 m buffer zone between them. Each main plot was divided into three replicates (23 m × 4 m each), and each replicate was divided into eight subplots (2 m × 4 m each), with a 1 m buffer zone between two adjacent subplots. Each CF subplot contained eight wheat rows with 20 cm spacing. Each RB subplot contained two raised beds, and each bed was approximately 20 cm high and contained four wheat rows with 20 cm spacing. Each RF subplot contained four ridges, and each ridge was 20 cm deep (from the bottom of the furrow to the top of ridge) and 50 cm away from the adjacent ridge (center–center) (Figure 2).

In the treatments mulched plastic film, the entire area of the subplot was manually covered with 0.12 mm-thick polyethylene film before planting. To sow the seeds, longitudinal incisions were made in the plastic film at 20 cm intervals in the CF and RB treatments, and in the slopes of the ridges in RF treatments (Figure 2). In the treatments mulched with wheat straw, air-dried wheat straw was chopped into 5–10 cm pieces and evenly distributed over the soil surface at a rate of 0.6 kg m$^{-2}$ after seedling emergence (Figure 2).

Seeds of the spring wheat cultivar Summit were sown manually at a rate of 150 kg ha$^{-1}$ on 11 December 2019 and 22 November 2020. All treatments were fertilized with 180, 90, and 100 kg ha$^{-1}$ of N, P$_2$O$_5$, and K$_2$SO$_4$, respectively. The full dose of P, half dose of K, and one-third dose of N were applied as basal fertilizers before sowing. Plants were fertilized again with one-third dose of N at the tillering stage, and the remaining half dose of K and one-third dose of N at the booting stage. Calcium superphosphate (18.5% P$_2$O$_5$), potassium sulfate (50% K$_2$O), and ammonium nitrate (33.5% N) were used as sources of P, K, and N fertilizers, respectively. Weeds were removed manually, and diseases were controlled chemically, when necessary. Plants in all treatments were harvested on 226 April 2020 and 11 April 2021.

*2.3. Irrigation Treatments*

The quantity of irrigation water required ($ET_c$) for the 1.00 ET treatment was estimated based on the reference evapotranspiration rate ($ET_o$) and crop coefficient ($K_c$) as follows:

$$ET_c = ET_o \times K_c \tag{1}$$

$ET_o$ was calculated according to the modified Penman–Monteith equation [39], while the $K_c$ values of spring wheat reported in the FAO-56 were used after correction, based on the values of specific wind speed and relative humidity at the study site. Based on this calculation, the total amount of irrigation water required for the 1.00 ET treatment was estimated at approximately 6500 m$^3$ ha$^{-1}$. Half of this amount (3250 m$^3$ ha$^{-1}$) was applied to the 0.50 ET treatment. A low-pressure surface irrigation system was used to apply the irrigation water. This system consisted of a main line (76 mm in diameter) that carried the irrigation water from the main water source to the subplots. This main line branched off into the sub-main hoses and was equipped with a manual control valve at each subplot to control the amount of water delivered to each subplot.

$ET$ (mm) from the top 1.0 m soil layer in each treatment was calculated using the following water-balance equation [40]:

$$ET = ET_c + P + C \pm \Delta S - D - R \tag{2}$$

where $ET_c$ is the total irrigation water applied (mm); $P$ is the total precipitation in the growing season (mm); $C$ is the upward flow of groundwater into the root zone, which could be negligible, considering the lack of capillary rise from the groundwater; $\Delta S$ is the difference between the SWC at sowing and that at harvest in the top 1.0 m soil layer (mm); $D$ is the downward drainage of water from the root zone (mm); $R$ is the surface runoff (mm), which was neglected in this study, because irrigation water was applied using the low-pressure surface irrigation system that controlled the amount of water delivered to each subplot.

*2.4. SWC*

Soil samples were collected from each subplot at varying depths (0–20, 20–40, and 40–60 cm) using a soil ferric auger before each irrigation in 1.00 ET and 0.50 ET treatments and at 1 week and 2 weeks after irrigation in 0.50 ET treatment. Gravimetric SWC was determined by drying the soil samples at 105 °C to a constant weight, and volumetric SWC (mm) was calculated by multiplying gravimetric SWC with the depth of soil sampling (20 cm) and the bulk density of the soil sample.

*2.5. Plant Parameters and WUE*

At 90 days after sowing, 10 plants were randomly sampled from each subplot, and oven-dried at 70 °C to a constant weight. Then, plants were weighed to determine the plant dry weight (PDW).

At maturity, 20 main spikes were randomly sampled from each subplot, and grain number per spike (GNPS) and 1000-grain weight (TGW) were measured. Then, plants were harvested from an area of 4.0 m$^2$ in each subplot, air-dried for 7 days, and weighed to measure the total biological yield (BY). Subsequently, spikes of the harvested plants were threshed, and grains were collected to record the final GY. Harvest index (HI) was calculated by dividing GY (ton ha$^{-1}$) with BY (ton ha$^{-1}$), and WUE was calculated by dividing GY (kg ha$^{-1}$) with ET (mm).

*2.6. Relationship of Yield with ET, WUE, SWC, and Ky*

Regression analysis was conducted to evaluate the relationship of PDW and GY with seasonal ET or WUE, and to develop the production functions of the relationship between the measured parameters and SWC at a depth of 0–60 cm. The relationship between relative reduction in PDW or GY $(1-Y_a/Y_m)$ and relative deficit in *ET* $(1-ET_a/ET_m)$ was evaluated using *Ky*, which was calculated using the following equation [41]:

$$\left(1 - \frac{Y_a}{Y_m}\right) = Ky\left(1 - \frac{ET_a}{ET_m}\right) \tag{3}$$

where $Y_a$ and $Y_m$ are the maximum and minimum PDW (g plant$^{-1}$) or GY (Kg ha$^{-1}$), respectively, and $ET_a$ and $ET_m$ are the corresponding maximum and minimum *ET* (mm).

*2.7. Data Analysis*

Values of the different parameters measured in each season were subjected to analysis of variance (ANOVA), appropriate for a split-plot design. Irrigation treatment, PP, and their interaction were considered as fixed effects, and replication was considered as a random effect. The significance of the difference between treatment means was tested by Duncan's multiple range test at the 5% probability level. The procedure was implemented utilizing the SPSS20.0 software (IBM Inc., Chicago, IL, USA). Linear and quadratic regression analyses were performed to examine the relationship between the different measured parameters and SWC at a depth of 0–60 cm. The different relationships were plotted using SigmaPlot (v. 11.0; SPSS, Chicago, IL, USA).

## 3. Results

*3.1. Soil Water Status*

Figure 3 shows the SWC distribution at depths of 0–20, 20–40, and 40–60 cm under the limited irrigation rate (0.50 ET); values were measured before irrigation (at 56, 87, and 108 days after planting), 7 days after irrigation (at 63, 94, and 115 days after planting), and 14 days after irrigation (at 70, 101, and 122 days after planting). The PP treatments had a significant effect on the SWC at different soil depths in the 0.50 ET treatment, with the highest SWC in the plastic mulch treatments (RFPM, CFPM, and RBPM) and lowest SWC in the non-mulched treatments (CFNM and RBNM) (Figure 3). Compared with the two non-mulched treatments, the mean SWC values in the three plastic mulch treatments were significantly higher all three time points (before irrigation, 7 days after irrigation,

and 14 days after irrigation): by 14.4–22.0%, 17.6–24.8%, and 7.0–18.7%, respectively, at 0–20 cm depth; by 16.4–29.0%, 18.4–29.1%, and 13.0–28.9%, respectively, at 20–40 cm depth; and by 3.3–34.8%, 4.9–32.7%, and 0.4–25.5%, respectively, at 40–60 cm depth. Compared with the two non-mulched treatments, the mean values of SWC in wheat straw mulch treatments (CFSM and RBSM) were significantly higher before irrigation, 7 days after irrigation, and 14 days after irrigation: by 6.9–17.2%, 3.1–17.4%, and 1.1–6.9%, respectively, at 0–20 cm depth; by 6.6–14.9%, 0.6–19.9%, and 3.2–20.2%, respectively, at 20–40 cm depth; and by 3.4–14.5%, 4.1–16.4%, and 1.1–16.5%, respectively, at 40–60 cm depth. The SWC of the RFNM treatment was occasionally comparable with those of non-mulched PPs at soil depths of 0–20 and 20–40 cm and of plastic film and wheat straw mulch treatments at a depth of 40–60 cm (Figure 3).

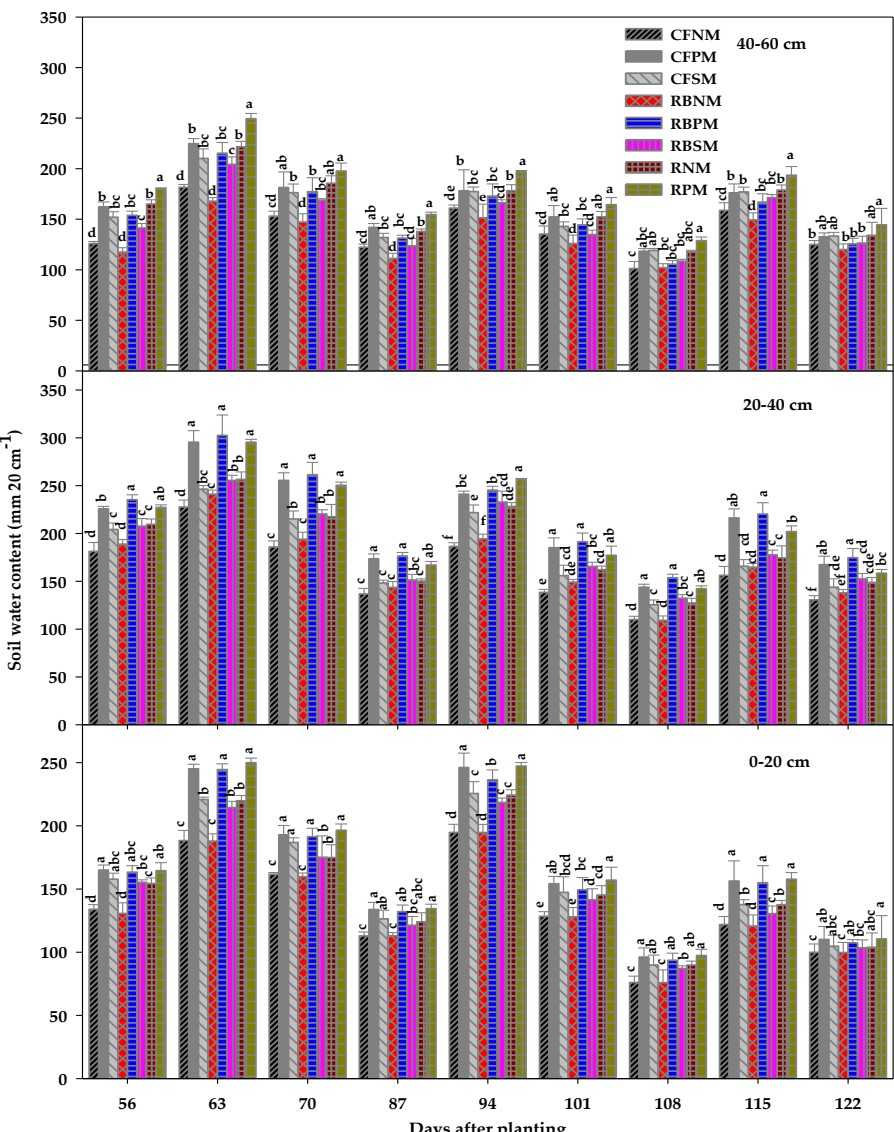

**Figure 3.** Effects of different planting patterns under limited irrigation rate (0.50 ET) on soil water storage at a 20 cm depth interval within 0–60 cm over two growing seasons at different durations (in days) after sowing. Conventional flat planting (CF) without mulching (CFNM); CF with plastic film mulch (CFPM); CF with wheat straw mulch (CFSM); raised-bed planting (RB) without mulch (RBNM); RB with plastic film mulch (RBPM); RB with wheat straw mulch (RBSM); ridge–furrow planting (RF) without mulch (RFNM); and RF with plastic film mulch on both ridges and furrows (RFPM). Bars with different letters are significantly different from each another on the basis of Duncan's multiple range test at $p \leq 0.05$.

Figure 4 shows the SWC measured prior to irrigation in the 1.00 ET plots at different soil depths. Compared with PPs mulched with either plastic sheet (RFPM, CFPM, and RBPM) or wheat straw (CFSM and RBSM), the two non-mulched PPs (CFNM and RBNM) displayed a reduction in SWC by 10.3–13.1%, 10.8–12.5%, and 10.0–14.7% at soil depths of 0–20, 20–40, and 40–60 cm, respectively. Additionally, the SWC of RFNM was comparable with those of non-mulched treatments at a depth of 0–20 cm and of wheat straw mulch treatments at soil depths of 20–40 and 40–60 cm (Figure 4).

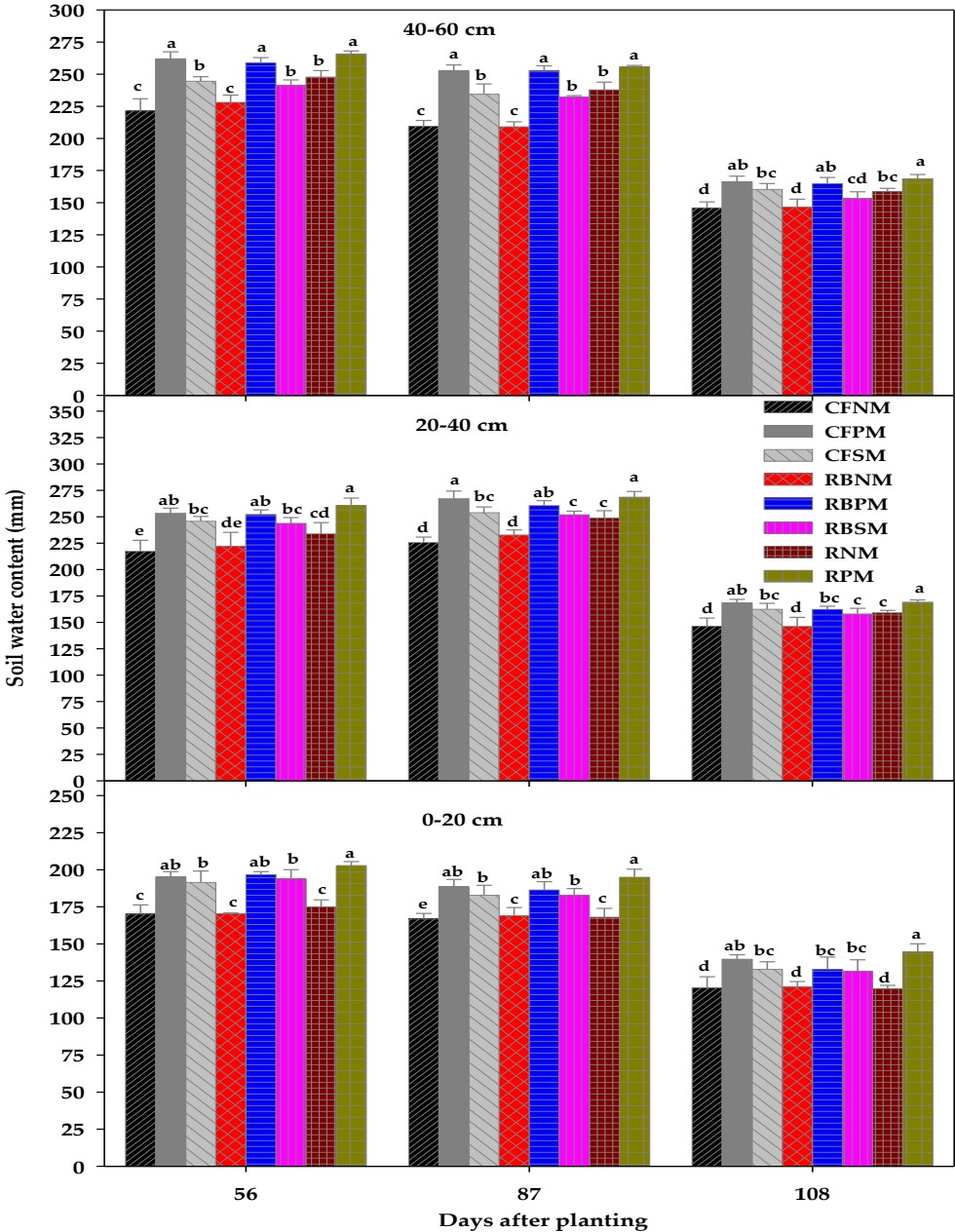

**Figure 4.** Effects of different planting patterns under full irrigation rate (1.00 ET) on soil water storage before irrigation at a 20 cm depth interval within 0–60 cm over two growing seasons at different durations (in days) after sowing. Conventional flat planting (CF) without mulching (CFNM); CF with plastic film mulch (CFPM); CF with wheat straw mulch (CFSM); raised-bed planting (RB) without mulch (RBNM); RB with plastic film mulch (RBPM); RB with wheat straw mulch (RBSM); ridge–furrow planting (RF) without mulch (RFNM); and RF with plastic film mulch on both ridges and furrows (RFPM). Bars with different letters are significantly different from each another on the basis of Duncan's multiple range test at $p \leq 0.05$.

The total SWC in the 0–60 cm soil layer in the 0.50 ET and 1.00 ET treatments is shown in Figure 5. In both water treatments, the PPs mulched with plastic sheet (RFPM, CFPM, and RBPM) showed the highest SWC, followed by PPs mulched with wheat straw (CFSM and RBSM) and RFNM, while the non-mulched PPs (CFNM and RBNM) showed the lowest SWC. In the 0.50 ET treatment, the PPs mulched with plastic sheet (RFPM, CFPM, and RBPM) and wheat straw (CFSM and RBSM) showed significantly higher total SWC in the 0–60 cm soil layer compared with non-mulched PPs (CFNM and RBNM) at all three time points: 17.5–21.5% and 8.7–14.9% higher, respectively, before irrigation; 19.6–22.9% and 9.9–13.3% higher, respectively, at 7 days after irrigation; and 13.1–21.1% and 7.0–12.7% higher, respectively, at 14 days after irrigation. In the 1.00 ET treatment, the PPs mulched with plastic sheet and wheat straw showed significantly higher total SWC (12.5–14.1% and 6.7–8.9%, respectively) in the 0–60 cm layer before irrigation compared with non-mulched PPs (Figure 5).

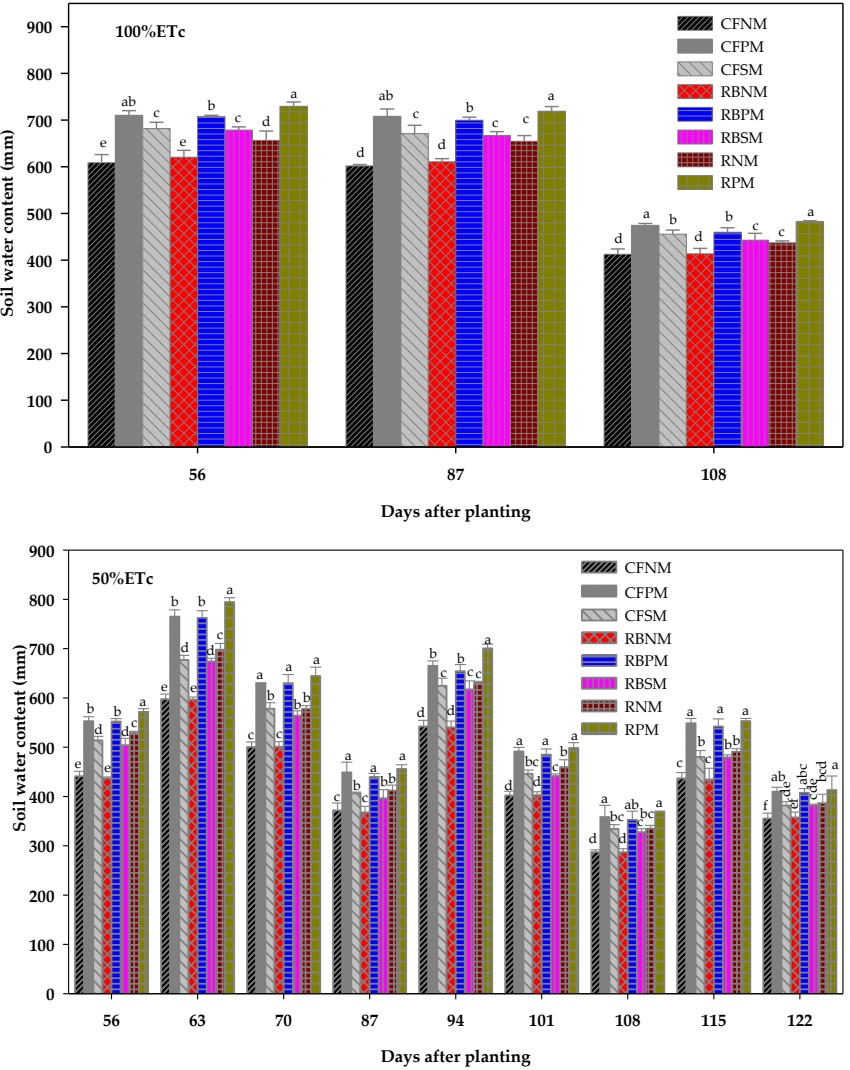

**Figure 5.** Effects of different planting patterns under full (1.00 ET) and limited (0.50 ET) irrigation rate on total soil water storage in the 0–60 cm soil layer over two growing seasons at different durations (in days) after sowing. Conventional flat planting (CF) without mulching (CFNM); CF with plastic film mulch (CFPM); CF with wheat straw mulch (CFSM); raised-bed planting (RB) without mulch (RBNM); RB with plastic film mulch (RBPM); RB with wheat straw mulch (RBSM); ridge–furrow planting (RF) without mulch (RFNM); and RF with plastic film mulch on both ridges and furrows (RFPM). Bars with different letters are significantly different from each another on the basis of Duncan's multiple range test at $p \leq 0.05$.

### 3.2. Growth, Yield and Yield Components, and WUE

The results of ANOVA (Tables 1 and 2) showed significant main effect ($p \leq 0.05$ and 0.01) of irrigation rate (I) and PPs on all measured parameters in the two growing seasons, except HI, which showed nonsignificant variation among the eight PPs. The I × PP interaction also had a significant effect on all measured parameters, except HI in the second season (Table 2).

**Table 1.** Plant dry weight (PDW), grain number spike–1 (GNPS), 1000-grain weight (TGW), grain yield (GY), biological yield (BY), harvest index (HI), and water use efficiency (WUE), as affected by different combination of planting patterns (PP) and mulching under full (1.00 ET) and limited (0.50 ET) irrigation rate (I), and as main effects for the first season.

| PP | PDW | GNPs | TGW | GY | BY | HI | WUE |
|---|---|---|---|---|---|---|---|
| | | | | **1.00 ET** | | | |
| CFNM | 9.47 ab | 53.14 bc | 40.80 b | 6.69 bc | 17.35 ab | 38.60 ab | 10.12 ab |
| CFPM | 9.56 ab | 54.34 b | 40.71 b | 6.99 b | 17.58 ab | 39.77 ab | 11.12 a |
| CFSM | 10.04 a | 58.71 a | 45.64 a | 7.65 a | 18.63 a | 41.09 a | 11.78 a |
| RBNM | 9.55 ab | 53.24 b | 40.62 b | 6.96 b | 17.40 ab | 40.08 ab | 10.52 ab |
| RBPM | 9.41 ab | 54.88 b | 40.45 b | 6.64 bc | 17.27 b | 38.46 ab | 10.48 ab |
| RBSW | 9.96 a | 58.54 a | 45.76 a | 7.56 a | 18.58 a | 40.85 a | 11.70 a |
| RFNM | 8.55 b | 49.66 c | 40.62 b | 6.23 c | 16.84 b | 36.97 b | 9.77 b |
| RFPM | 8.48 b | 50.38 c | 40.90 b | 6.41 c | 16.81 b | 38.14 ab | 10.27 ab |
| Mean | **9.38 A** | **54.11 A** | **41.94 A** | **6.89 A** | **17.56 A** | **39.24 A** | **10.72 B** |
| | | | | **0.50 ET** | | | |
| CFNM | 4.90 cd | 35.60 bc | 29.19 bc | 3.21 cd | 11.84 bcd | 27.13 c | 8.18 ef |
| CFPM | 7.56 a | 42.80 ab | 33.37 ab | 4.81 ab | 14.35 a | 33.49 ab | 15.48 a |
| CFSM | 6.62 abc | 37.83 bc | 31.29 bc | 3.65 cd | 12.49 bc | 29.23 bc | 10.41 d |
| RBNM | 4.77 d | 34.77 c | 28.79 c | 3.14 d | 11.36 d | 27.71 c | 7.95 f |
| RBPM | 6.98 ab | 41.13 ab | 32.86 ab | 4.31 b | 14.05 a | 30.73 bc | 13.48 b |
| RBSW | 6.40 abc | 37.47 bc | 29.55 bc | 3.44 cd | 11.65 cd | 29.55 bc | 9.73 de |
| RFNM | 6.33 bc | 38.33 bc | 32.53 b | 3.83 c | 12.78 b | 30.10 bc | 11.97 c |
| RFPM | 7.54 a | 43.67 a | 34.57 a | 5.06 a | 14.25 a | 35.48 a | 16.31 a |
| Mean | **6.39 B** | **38.95 B** | **31.52 B** | **3.93 B** | **12.85 B** | **30.43 B** | **11.69 A** |
| | | | | F test | | | |
| I | ** | *** | * | ** | *** | ** | * |
| PP | * | * | * | *** | ** | ns | *** |
| I × PP | ** | ** | *** | *** | *** | * | *** |

Means in column followed by the same lowercase letter are not significantly different between eight PP at $p = 0.05$ according to Duncan's multiple range test. Means in column followed by the same uppercase letter are not significantly different between two irrigation rates at $p = 0.05$ according to Duncan's multiple range test. *,**, and *** indicate significant at $p < 0.05$, 0.01, and 0.001, respectively; ns indicates not significant. Conventional flat planting (CF) without mulching (CFNM); CF with plastic film mulch (CFPM); CF with wheat straw mulch (CFSM); raised-bed planting (RB) without mulch (RBNM); RB with plastic film mulch (RBPM); RB with wheat straw mulch (RBSM); ridge–furrow planting (RF) without mulch (RFNM); and RF with plastic film mulch on both ridges and furrows (RFPM).

Compared with the 1.00 ET treatment, the values of PDW, GNPS, TGW, GY, BY, and HI decreased by 31.9%, 28.0%, 24.8%, 43.0%, 26.8%, and 22.5%, respectively, in the first season (Table 1) and by 31.1%, 24.9%, 18.1%, 36.1%, 27.5%, and 12.1%, respectively, in the second season (Table 2) in the 0.50 ET treatment. However, WUE values were higher by 8.3% and 14.9% in the first and second seasons, respectively, under 0.05 ET than under 1.00 ET (Tables 1 and 2).

Regardless of the irrigation rate, the different PPs affected all parameters, except HI (Figure 6). The non-mulched conventional flat (CFNM) and raised-bed (RBNM) PPs always exhibited the lowest values of all parameters, whereas the corresponding PPs mulched with plastic sheet (CFPM and RBPM) and wheat straw (CFSM and RBSM) showed the highest values of all parameters (Figure 6). The average values of PDW, GNPS, TGW,

GY, BY, HI, and WUE over the two seasons were significantly increased by 12.0–16.5%, 8.1–10.0, 5.8–8.9%, 8.1–12.1%, 6.5–8.8%, 1.5–5.4%, and 11.7–26.1%, respectively, in these four PPs (CFPM, RBPM, CFSM, and RBSM) compared with either CFNM or RBNM (Figure 6). Values of all parameters in the ridge–furrow treatment mulched with plastic film (RFPM) were comparable to those in PPs mulched with plastic film (CFPM and RBPM) or wheat straw (CFSM and RBSM). In the RFPM treatment, values of PDW, GNPS, TGW, GY, BY, HI, and WUE were higher by 10.6%, 6.5%, 6.8%, 9.6%, 5.7%, 6.1%, and 26.1, respectively, compared with their values in the CFNM or RBNM treatment. Although the RFNM did not use mulch, the PDW, GNPS, TGW, BY, and WUE values in this treatment were 4.4%, 1.2%, 4.2%, 1.7%, and 12.2% higher, respectively, than those in the CFNM or RBNM treatment (Figure 6).

**Table 2.** Plant dry weight (PDW), grain number spike–1 (GNPS), 1000-grain weight (TGW), grain yield (GY), biological yield (BY), harvest index (HI), and water use efficiency (WUE), as affected by different combination of planting patterns (PP) and mulching under full (1.00 ET) and limited (0.50 ET) irrigation rate (I), and as main effects for the second season.

| PP | PDW | GNPs | TGW | GY | BY | HI | WUE |
|---|---|---|---|---|---|---|---|
| | | | | **1.00 ET** | | | |
| CFNM | 9.86 abc | 53.17 bc | 41.03 b | 6.78 bcd | 18.30 cde | 37.09 a | 10.31 bc |
| CFPM | 9.98 ab | 54.77 b | 41.82 b | 6.89 bc | 19.49 bc | 35.40 a | 10.97 bc |
| CFSM | 10.56 ab | 58.25 a | 46.11 a | 7.77 a | 20.96 a | 37.12 a | 11.99 a |
| RBNM | 9.82 bc | 53.08 bc | 41.15 b | 6.60 bcd | 18.27 de | 36.16 a | 9.98 cd |
| RBPM | 9.80 bc | 54.57 b | 42.91 b | 7.00 b | 19.01 cd | 36.83 a | 11.12 ab |
| RBSW | 10.62 a | 58.61 a | 46.25 a | 7.49 a | 20.43 ab | 36.70 a | 11.43 a |
| RFNM | 8.82 c | 50.95 c | 41.45 b | 6.26 d | 17.67 e | 35.45 a | 9.66 d |
| RFPM | 8.91 c | 50.73 c | 41.46 b | 6.42 cd | 17.77 e | 36.18 a | 10.27 bc |
| Mean | 9.80 A | 54.27 A | 42.77 A | 6.90 A | 18.99 A | 36.37 A | 10.72 B |
| | | | | 0.50 ET | | | |
| CFNM | 5.12 d | 36.25 bc | 33.09 ab | 4.06 b | 12.89 bc | 31.49 a | 10.41 d |
| CFPM | 8.06 a | 45.43 a | 37.24 a | 4.91 a | 14.48 ab | 33.94 a | 15.02 ab |
| CFSM | 6.58 bc | 38.81 bc | 34.47 ab | 4.10 b | 13.68 bc | 29.95 a | 11.20 d |
| RBNM | 5.19 d | 35.96 c | 32.48 b | 4.04 b | 12.76 c | 31.64 a | 10.25 d |
| RBPM | 7.84 a | 45.02 a | 36.14 a | 4.74 a | 14.05 ab | 33.73 a | 14.23 b |
| RBSW | 6.37 c | 38.59 bc | 34.30 ab | 4.07 b | 13.59 bc | 29.93 a | 11.13 d |
| RFNM | 6.99 b | 40.75 b | 35.32 a | 4.33 b | 13.82 b | 31.30 a | 12.83 c |
| RFPM | 7.88 a | 45.19 a | 37.14 a | 5.05 a | 14.91 a | 33.85 a | 15.77 a |
| Mean | 6.75 B | 40.75 B | 35.02 B | 4.41 B | 13.77 B | 31.98 B | 12.60 A |
| | | | | F test | | | |
| I | ** | ** | * | *** | ** | * | ** |
| PP | *** | *** | * | ** | *** | ns | *** |
| I × PP | *** | *** | * | *** | *** | ns | *** |

Means in column followed by the same lowercase letter are not significantly different between eight PP at *p* = 0.05 according to Duncan's multiple range test. Means in column followed by the same uppercase letter are not significantly different between two irrigation rates at *p* = 0.05 according to Duncan's multiple range test. *,**, and *** indicate significant at *p* < 0.05, 0.01, and 0.001, respectively; ns indicates not significant. Conventional flat planting (CF) without mulching (CFNM); CF with plastic film mulch (CFPM); CF with wheat straw mulch (CFSM); raised-bed planting (RB) without mulch (RBNM); RB with plastic film mulch (RBPM); RB with wheat straw mulch (RBSM); ridge–furrow planting (RF) without mulch (RFNM); and RF with plastic film mulch on both ridges and furrows (RFPM).

The response of different parameters to the different PPs varied with the irrigation rate. Under 1.00 ET, the highest values of all parameters were obtained in the two PPs mulched with wheat straw (CFSM and RBSM), followed by conventional flat and raised-bed PPs mulched with plastic film (CFPM and RBPM) or those without mulch (CFNM and RBNM), while the both ridge–furrow PPs (RFNM and RFPM) exhibited the lowest values for most parameters in both growing seasons (Tables 1 and 2). Under 0.50 ET, the highest values of

all parameters were observed with the three PPs mulched with plastic film (CFPM, RBPM, and RFPM), while the lowest values were obtained in the non-mulched PPs (CFNM and RBNM). The values of most parameters were comparable between PPs mulched with wheat straw (CFSM and RBSM) and those without mulch (CFNM and RBNM) (Tables 1 and 2).

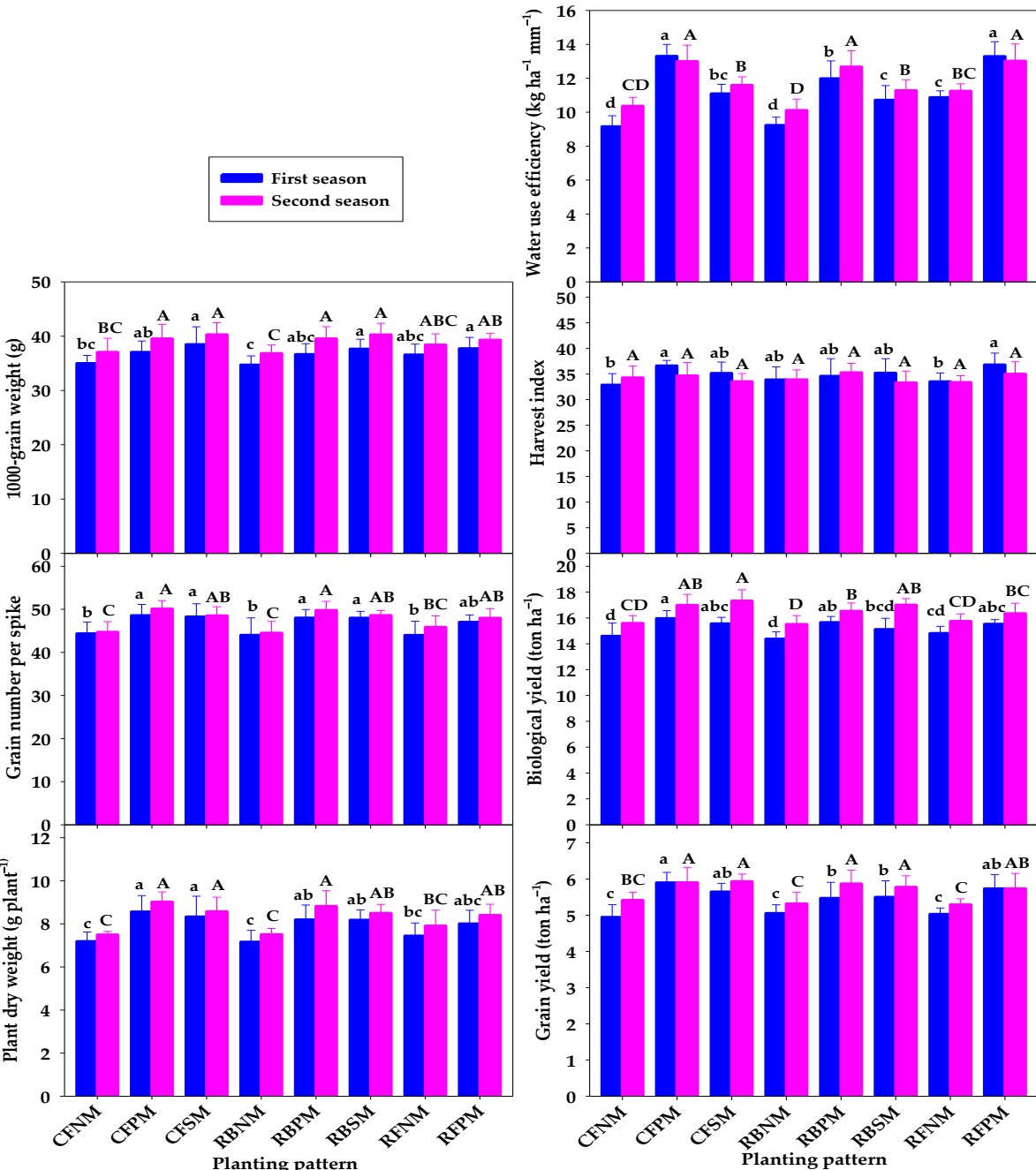

**Figure 6.** Effects of different combination of planting patterns (PP) and mulching on all measured parameters in the first and second seasons. Conventional flat planting (CF) without mulching (CFNM); CF with plastic film mulch (CFPM); CF with wheat straw mulch (CFSM); raised-bed planting (RB) without mulch (RBNM); RB with plastic film mulch (RBPM); RB with wheat straw mulch (RBSM); ridge–furrow planting (RF) without mulch (RFNM); and RF with plastic film mulch on both ridges and furrows (RFPM). Means followed by the same lowercase letter are not significantly different between eight PP in the first season at *p* = 0.05 according to Duncan's multiple range test. Means followed by the same uppercase letter are not significantly different between eight PP in the second season at *p* = 0.05 according to Duncan's multiple range test.

### 3.3. Measurement of Ky

Figure 7 shows the Ky data pooled for I and PP treatments and its relationship with the relative decrease in PDW ($Ky_{PDW}$) at 90 days after sowing or final GY ($Ky_{GY}$) and the corresponding relative ET deficits. The values (slopes) of Ky in the first and second seasons were less than 1.0 for both $Ky_{PDW}$ (0.61 and 0.56, respectively; Figure 7A) and $Ky_{GY}$ (0.70 and 0.67, respectively; Figure 7B).

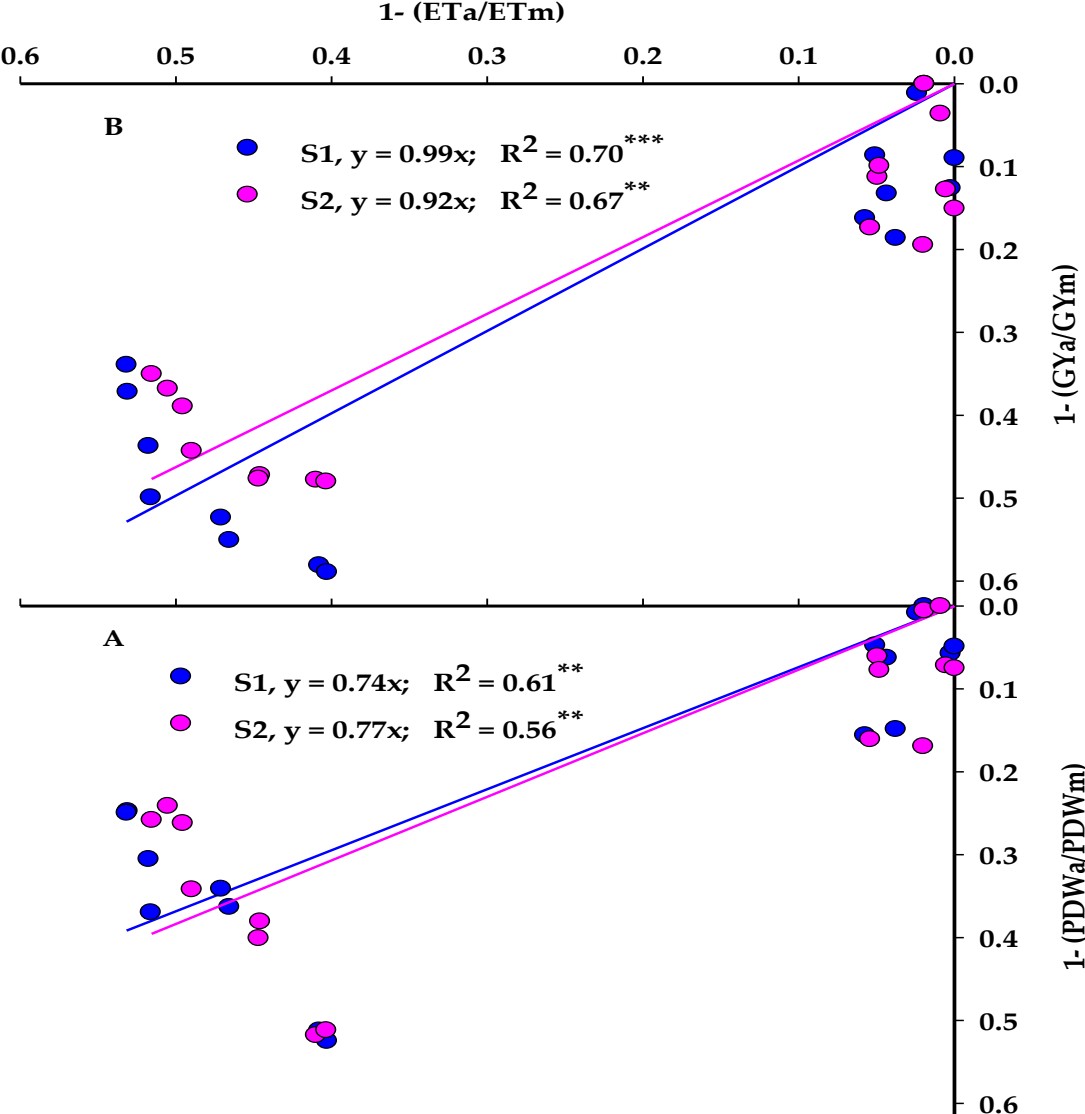

**Figure 7.** Yield response factor (ky) shown as the relationship between the relative plant dry weight decrease PDW (**A**) or grain yield decrease GY (**B**) and the corresponding relative evapotranspiration (ET) decrease for first season (S1) and second season (S2). **, and *** indicate significant at $p < 0.01$ and 0.001, respectively.

The different PPs had significant impacts on $Ky_{PDW}$ and $Ky_{GY}$ under 0.50 ET, with the respective values for the three PPs mulched with plastic sheet (CFPM, RBPM, and RFPM) and for the ridge–furrow PP without mulching (RFNM) being less than 1.0 in both seasons (Table 3). Values of $Ky_{PDW}$ for PPs mulched with wheat straw (CFSM and RBSM) were less than 1.0, where those of $Ky_{GY}$ for both treatments were slightly higher than 1.0. Values of $Ky_{PDW}$ and $Ky_{GY}$ for the non-mulched PPs (CFNM and RBNM) were significantly higher than 1.0 (Table 3).

**Table 3.** Yield response factor (ky) shown as the relationship between the relative biomass decrease ($Ky_{PDW}$) and relative grain yield decrease ($Ky_{GY}$) and the corresponding relative evapotranspiration for different combination of planting patterns (PP) and mulching under 0.50 ET treatment in two growing seasons.

| PP | First Season | | Second Season | |
|---|---|---|---|---|
| | $Ky_{PDW}$ | $Ky_{GY}$ | $Ky_{PDW}$ | $Ky_{GY}$ |
| CFNM | 1.25 | 1.42 | 1.26 | 1.16 |
| CFPM | 0.47 | 0.70 | 0.48 | 0.73 |
| CFSM | 0.72 | 1.11 | 0.85 | 1.06 |
| RBNM | 1.30 | 1.46 | 1.27 | 1.19 |
| RBPM | 0.59 | 0.84 | 0.53 | 0.79 |
| RBSW | 0.78 | 1.18 | 0.90 | 1.07 |
| RFNM | 0.72 | 0.97 | 0.70 | 0.90 |
| RFPM | 0.47 | 0.64 | 0.50 | 0.68 |

Conventional flat planting (CF) without mulching (CFNM); CF with plastic film mulch (CFPM); CF with wheat straw mulch (CFSM); raised-bed planting (RB) without mulch (RBNM); RB with plastic film mulch (RBPM); RB with wheat straw mulch (RBSM); ridge–furrow planting (RF) without mulch (RFNM); and RF with plastic film mulch on both ridges and furrows (RFPM).

### 3.4. Relationship of Yield with ET and WUE

The relationships of PDW and GY with ET in both growing seasons are shown in Figure 8. ET displayed a linear relationship with PDW and GY, and explained 67% and 61% of the variation in PDW (Figure 8A) and 78% and 83% of the variation in GY (Figure 8B) in the first and second seasons, respectively. The regression slope predicted an increase in PDW of 0.092 and 0.094 g plant$^{-1}$ (Figure 8A) and an increase in GY of 90.3 and 80.2 kg ha$^{-1}$ (Figure 8B) in the first and second seasons, respectively, for each 10 mm increase in ET. Additionally, the basal amount of ET required to start the production of GY was determined to be 90.4 mm in the first season and 189.8 mm in the second season (140.1 mm on average) (Figure 8B).

The polynomial model was the best to describe the relationship of WUE with PDW and GY (Figure 8). This relationship was significant for PDW ($R^2$ = 0.52 and 0.43; Figure 8C) but not significant for GY ($R^2$ = 0.33 and 0.11; Figure 8D) in the first and second seasons, respectively. Additionally, this relationship shows that WUE increased with the increase in PDW until the PDW values reached 7.43 g plant$^{-1}$ in the first season and 7.86 g plant$^{-1}$ in the second season; subsequently, WUE decreased with the increase in PDW (Figure 8C). The value of WUE also increased with the increase in GY until reaching 5000 kg ha$^{-1}$ in both seasons; subsequently, WUE decreased with the increase in GY (Figure 8D).

### 3.5. Production Function

Next, we evaluated the functional relationship between SWC measured prior to the next irrigation at different soil depths and the value of each wheat parameter in both irrigation treatments (Figures 9 and 10). Under 1.00 ET (Figure 9), the response function of different measured wheat parameters to SWC measured before irrigation (at 56, 87, and 108 days after planting) at three different soil depths (0–20, 20–40, and 40–60 cm) was quadratic. Additionally, the SWC showed a weak and nonsignificant relationship with all parameters, with the exception of the amount of SWC at 108 days after planting at a depth of 0–20 cm, which exhibited a strong relationship with almost all measured parameters ($R^2$ range = 0.61 to 0.73) (Figure 9). Under 0.50 ET (Figure 10), the production functions of different measured wheat parameters vs. the amount of SWC at different soil depths and measurement time points were linear, with the exception of GY, BY, HI, and WUE, which showed a quadratic relationship with the amount of SWC either at 0–20 cm depth (GY, BY, and WUE) or at 40–60 cm depth (HI). Additionally, the amount of SWC at different soil depths and measurement time points exhibited a strong relationship with all measured wheat parameters ($R^2$ range = 0.68–0.99), with the exception of the amount of

SWC at 108 days after planting at a depth of 40–60 cm, which showed a weak–moderate relationship ($R^2$ range = 0.44 to 0.60) with all measured wheat parameters (Figure 10).

Table 4 shows the relationship between the different measured wheat parameters and the total SWC in the 0–60 cm soil layer measured prior to the next irrigation. All parameters showed a linear relationship with total SWC in the 0–60 cm soil layer at all three time points, with exception of WUE, which showed a second order relationship. Total SWC in the 0–60 cm soil layer showed strong relationships with all measured wheat parameters ($R^2$ range = 0.82 to 0.90) at all three time points, except WUE, which showed a weak and nonsignificant relationship (Table 4). Based on the regression slope of these relationships, the increase in PDW, GNPS, TGW, GY, BY, HI, and WUE ranged from 0.117 to 0.249 g per plant, from 0.548 to 1.138 grains per spike, from 0.347 to 0.715 g per TGW, from 0.103 to 0.212 ton ha$^{-1}$, 0.188 to 0.387 ton ha$^{-1}$, from 0.253 to 0.526%, and from 50.1 to 90.2 kg ha$^{-1}$, respectively, for each 10 mm increase in SWC in the 0–60 cm soil layer (Table 4).

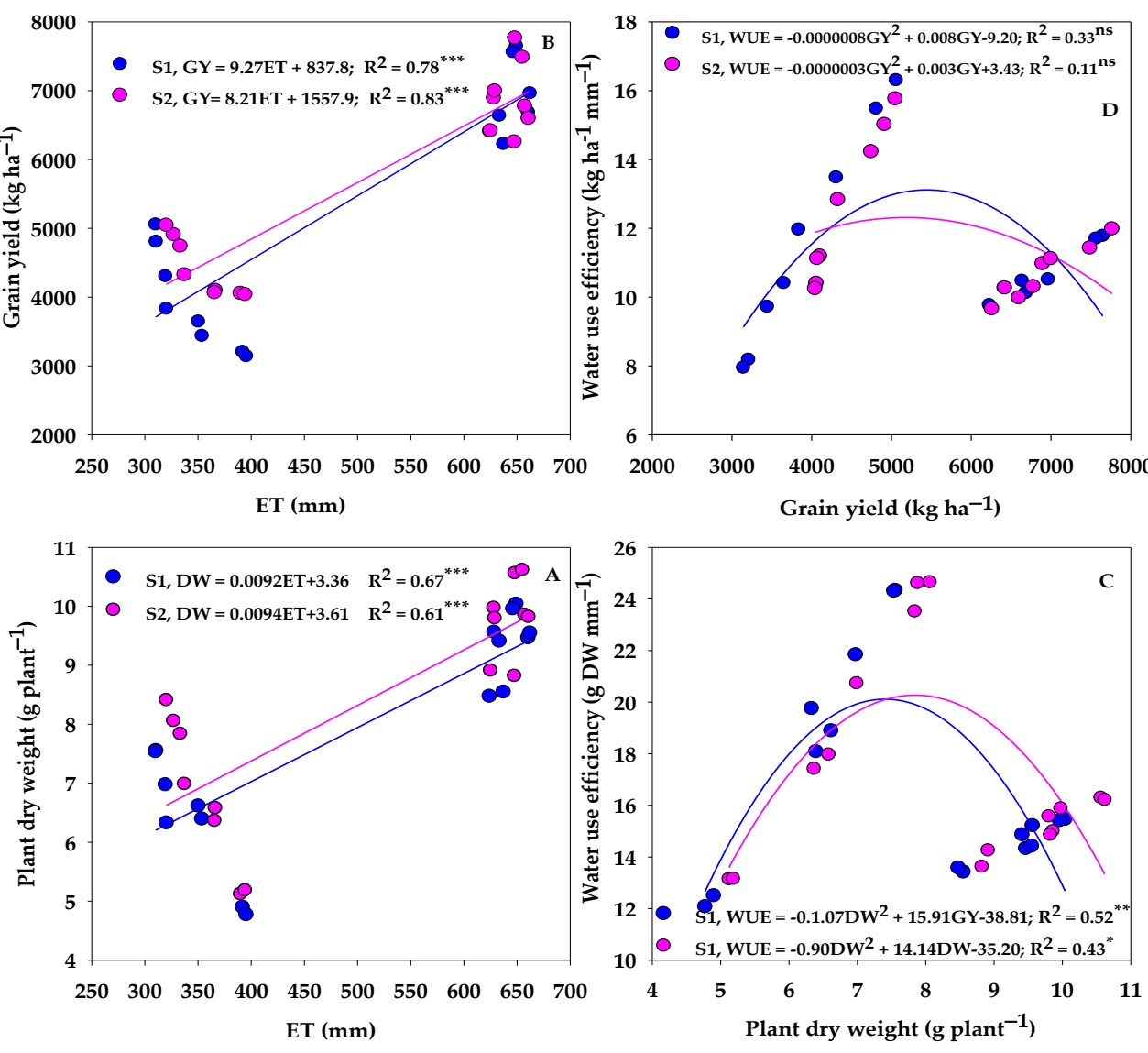

**Figure 8.** Relationship of seasonal evapotranspiration (ET) with plant dry weight (**A**) and grain yield (**B**) as well as relationship of water use efficiency with plant dry weight (**C**) and grain yield (**D**) in the first season (S1) and second season (S2). *,**, and *** indicate significant at *p* < 0.05, 0.01, and 0.001, respectively; ns indicates not significant.

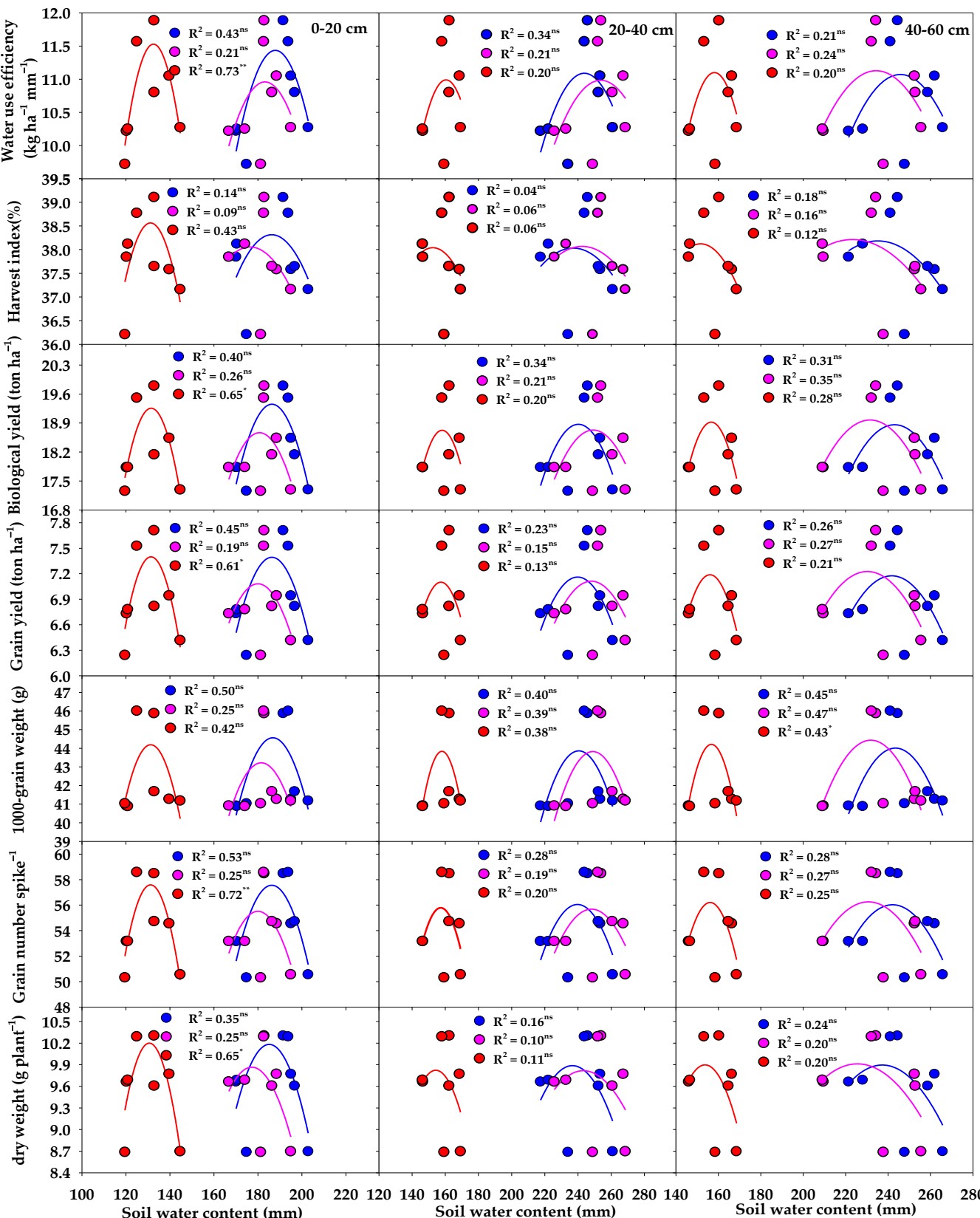

**Figure 9.** Functional relationship between different plant parameters and the soil water content measured before irrigation at 56 (circle green), 87 (circle pink), and 108 (circle red) days after planting at deferent soil depth under full irrigation rate (1.00 ET). *, and ** indicate significant at $p < 0.05$ and 0.01, respectively; ns indicates not significant.

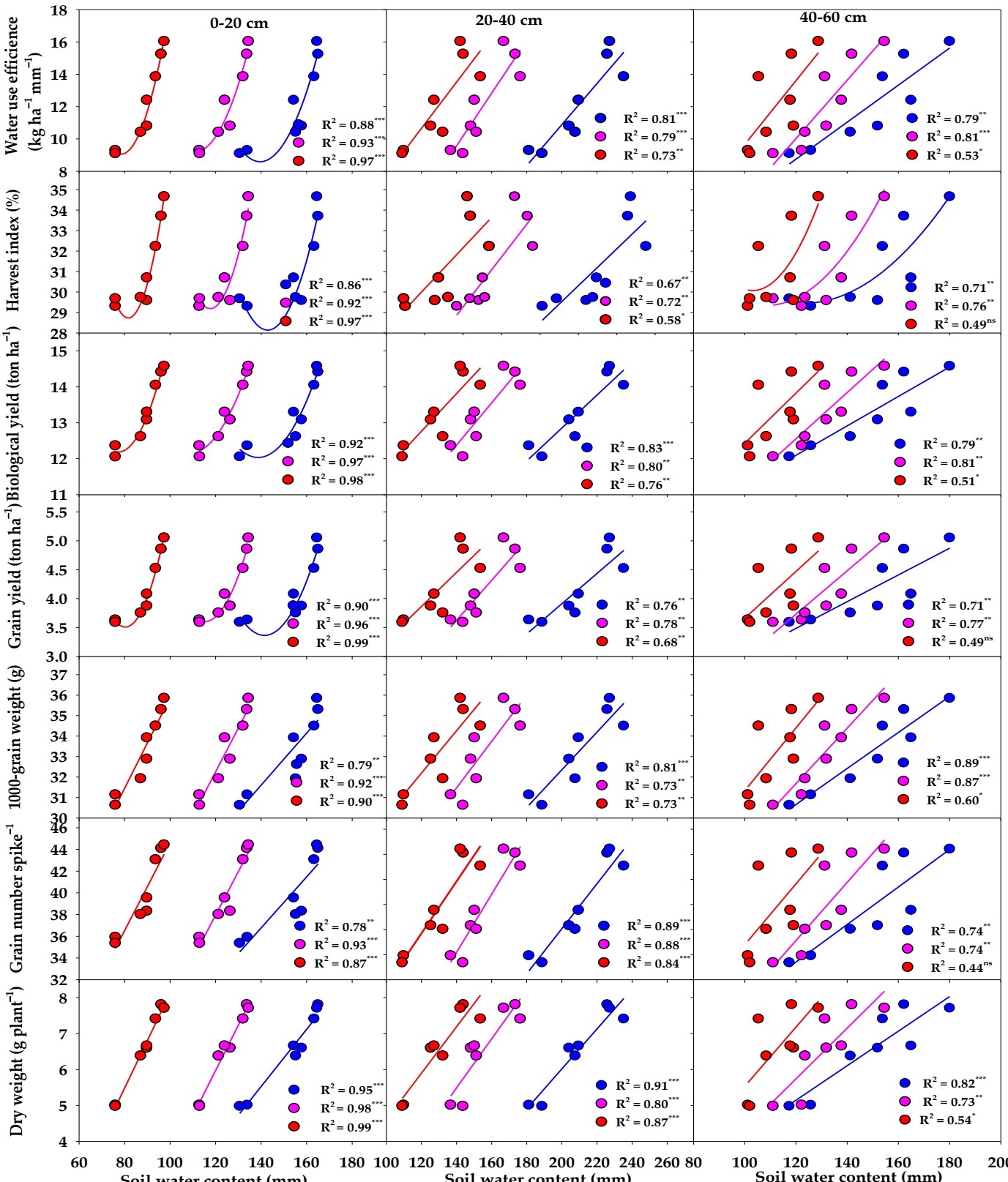

**Figure 10.** Functional relationship between different plant parameters and the soil water content measured before irrigation at 56 (circle green), 87 (circle pink), and 108 (circle red) days after planting at deferent soil depth under limited irrigation rate (0.50 ET). *,**, and *** indicate significant at $p < 0.05$, 0.01, and 0.001, respectively; ns indicates not significant.

**Table 4.** Functional relationship between different measured wheat parameters (Y) and total soil water storage in the 0–60 cm soil layer (x) measured before irrigation.

| Parameters | Days after Sowing | Equation | $R^2$ | Increase in Each Parameters for Each 10 mm Increase in SWS |
|---|---|---|---|---|
| Plant dry weight (g plant$^{-1}$) | 56 | Y = 0.0171x − 2.05 | 0.83 | 0.171 g per plant |
| | 87 | Y = 0.0117x + 1.77 | 0.81 | 0.117 g per plant |
| | 108 | Y = 0.0249x − 1.59 | 0.84 | 0.249 g per plant |
| Grain number per spike | 56 | Y = 0.0779x + 0.78 | 0.84 | 0.779 grain per spike |
| | 87 | Y = 0.0548x + 17.39 | 0.87 | 0.548 grain per spike |
| | 108 | Y = 0.1138x + 2.77 | 0.85 | 1.138 grain per spike |
| 1000-grain weight (g) | 56 | Y = 0.0491x + 8.68 | 0.83 | 0.491 g per 1000-grain |
| | 87 | Y = 0.0347x + 19.08 | 0.87 | 0.347 g per 1000-grain |
| | 108 | Y = 0.0715x + 10.00 | 0.84 | 0.715 g per 1000-grain |
| Grain yield (ton ha$^{-1}$) | 56 | Y = 0.0144x − 3.02 | 0.83 | 0.144 ton ha$^{-1}$ |
| | 87 | Y = 0.0103x − 0.039 | 0.89 | 0.103 ton ha$^{-1}$ |
| | 108 | Y = 0.0212x − 2.71 | 0.87 | 0.212 ton ha$^{-1}$ |
| Biological yield (ton ha$^{-1}$) | 56 | Y = 0.0263x + 0.166 | 0.84 | 0.263 ton ha$^{-1}$ |
| | 87 | Y = 0.0188x + 5.63 | 0.90 | 0.188 ton ha$^{-1}$ |
| | 108 | Y = 0.0387x − 0.751 | 0.87 | 0.387 ton ha$^{-1}$ |
| Harvest index (%) | 56 | Y = 0.0358x + 13.27 | 0.82 | 0.358% |
| | 87 | Y = 0.0253x + 20.85 | 0.86 | 0.253% |
| | 108 | Y = 0.0526x + 14.05 | 0.85 | 0.526% |
| Water use efficiency (kg ha$^{-1}$ mm$^{-1}$) | 56 | Y = −0.00015x$^2$ − 0.0018x + 13.8 | 0.34 | 60.0 kg ha$^{-1}$ |
| | 87 | Y = −0.00012x$^2$ − 0.0022x + 14.72 | 0.32 | 90.2 kg ha$^{-1}$ |
| | 108 | Y = −0.00035x$^2$ − 0.0036x + 38.12 | 0.34 | 50.1 kg ha$^{-1}$ |

## 4. Discussion

### 4.1. Effects of PPs and Mulching Practices on SWC

Confronting the water crisis in arid and semiarid regions constantly requires innovative water-saving measures for promoting the effective use of water in the agriculture sector. Several studies have reported that crop productivity and irrigation WUE can be improved under limited irrigation water supply by reducing the amount of water that evaporates directly from the soil surface and has little utility for crop production; this amount of water accounts for up to 40% of the total crop water use [3,31–33,42–44]. It is likely that the combination between PP and mulching practices, the two variables that have the greatest impact on soil water storage, as well as soil water distribution within the root zone, soil temperature, and the amount of water lost from the soil surface through evaporation, could enhance crop growth and maximize crop yield and WUE simultaneously under limited water availability. Several studies on wheat have shown that in comparison with conventional flat planting without mulching (CFNM), ridge–furrow planting with plastic film or wheat straw mulching decreased ET, thus increasing the SWC and water availability over a relatively long period of time during the crop growing season [14,15,23,33,45,46]. Consequently, in this study, the SWC at different soil depths in the PPs mulched with plastic sheet (RFPM, CFPM, and RBPM) or wheat straw (CFSM and RBSM) was significantly higher by 0.4–34.8% and 0.6–20.2%, respectively, than in the two non-mulched PPs (CFNM and RBNM) under 0.50 ET (Figure 3). This range in SWC depends on the soil depth and the time of measurement relative to the day of irrigation (Figure 3). Under 1.00 ET, the two non-mulched PPs displayed a reduction in SWC by 10–13.7% compared with the different mulched PPs (Figure 4). In general, the results of this study highlight the importance of mulching practices as a water conservation strategy in water-scarce regions, with the plastic sheet mulch being more effective in conserving soil water than wheat straw mulch, particularly under limited water supply, and the latter being more effective than

no mulch. Additionally, ridge–furrow planting with plastic sheet mulching (RFPM) is a good strategy for conserving soil water under the limited irrigation rate. These findings are consistent with previous studies [15,47,48], which showed that the SWC was higher under plastic sheet mulch than under straw mulch. This may be because plastic film has the ability to block water from evaporating from the soil into the air (see Figure 2), thus greatly reducing soil evaporation, which in turn conserves more water in the soil [31,49]. The ability of wheat straw to conserve soil water may result from its capacity to decrease the soil temperature and latent heat flux, thus reducing soil evaporation [43,50]. Additionally, plant straw is a good material for absorbing and conserving irrigation water [20].

*4.2. The Impacts of PPs and Mulching Practices on Growth, Yield, and WUE Depending on Irrigation Rate*

An important finding of this study was the strong response of growth, yield components, GY, and WUE of spring wheat under arid conditions to the combination of PP and mulching practice. However, this response varied with the irrigation rate. Under 1.00 ET, the PPs mulched with wheat straw (CFSM and RBSM) performed better than the ridge–furrow PP with or without mulch (RFNM and RFPM). However, the three PPs mulched with plastic film (CFPM, RBPM, and RFPM) seemed to be superior in enhancing crop growth, production, and WUE under 0.50 ET compared with the other PPs, particularly those without mulch (CFNM and RBNM) (Tables 1 and 2). For instance, the average values of PDW, GY, and WUE over two seasons and the two wheat straw mulch treatments (CFSM and RBSM) under 1.00 ET were higher by approximately 15.7%, 18.0%, and 17.0%, respectively, than their values in RFNM or RFPM. However, under 0.50 ET, values of PDW, GY, and WUE in CFPM, RBPM, and RFPM treatments were approximately 34.5%, 24.5%, and 38.0% higher, respectively, than those in the CFNM or RBNM treatment (Tables 1 and 2). These results indicate that wheat straw mulching appears to be a useful practice when the amount of irrigation water is sufficient; however, the stimulatory impact of this practice on crop growth, yield, and WUE decreases with decreasing irrigation rate. The opposite result was found in the RFPM treatment, which was effective under the limited irrigation rate but failed to compete with the other PPs under the full irrigation rate, regardless of the mulching practice. Additionally, under limited irrigation rate and arid conditions, plastic sheet mulching was more effective than wheat straw mulching in mitigating the negative impacts of soil water limitation on wheat growth, which in turn enhanced the yield components, GY, and WUE. Furthermore, irrespective of mulching and PP types, the growth, yield components, GY, and WUE of wheat were significantly higher in mulched plots than in non-mulched plots under the limited irrigation rate. These findings can be explained as follows. First, straw mulching not only improves water availability by reducing soil evaporation but also improves several physiochemical and biological proprieties of soil [19,29,51–55]. In addition, the decomposition of wheat straw during the growing season leads to the slow release of nutrients, thereby preventing nutrient leaching and providing an additional source for plant nutrition [19,56,57]. Because most of the abovementioned advantages of wheat straw depend on its decomposition rate, which is always faster under high SWC than under low SWC [58,59], this might explain why the treatments mulched with wheat straw (CFSM and RBSM) were more effective than other treatments in enhancing the growth, production, and WUE of wheat under 1.00 ET. Second, in the RFPM treatment, the use of plastic film to cover the ridge and cultivation of plants in the furrow acted to concentrate the limited amount of irrigation water applied within the furrow and plant root zone, which helped maintain an optimal amount of SWC in the root zone [21,24]. Additionally, the furrow where the plants are grown receives less solar energy, because of the shade of the ridge and plants (see Figure 2), which decreases the rate of evaporation from the furrow and allows the root to reach deeper in the soil, thus promoting water uptake from the deeper soil layers [60,61]. However, on the other hand, in the RFPM system, only the 50 cm wide ridge was used for plant cultivation, and thus a significant area of the field was not utilized. Zhao et al. [19] and Zhou et al. [62] have reported that wheat

cultivation area under ridge–furrow planting is 24% less than that under conventional flat planting, which results in 23% less spikes. All the aforementioned RFPM-related facts might explain why this PP was very effective under 0.50 ET but failed to compete with any of the other PPs under 1.00 ET. Third, the growth, GY, and WUE of wheat under 0.50 ET were significantly higher in PPs mulched with the plastic sheet than in those mulched with wheat straw, probably because the plastic film serves as a more effective insulation layer than wheat straw and blocks the passage of water from the mulched soil to the atmosphere, collecting this water in the gap between the plastic film and the soil surface (see Figure 2). Therefore, this mechanism keeps the topsoil water content relatively stable and enables the movement of water from the deeper layers of soil to the topsoil through both vapor transfer and capillary action [63,64]; this might explain why PPs mulched with plastic film were more effective than those mulched with wheat straw under the limited irrigation rate.

*4.3. Identify the Optimal Coupling Combinations between PPs and Mulching Practices for Enhancing Growth, GY, and WUE under Limited Water Applications*

Identification of the best PP and mulching practice combination is crucial for enhancing the growth, yield, and WUE of wheat, especially when the water supply is limited. This objective can be achieved by calculating Ky, which is a proportionality factor between the relative loss of PDW or GY and relative water supply decrease [41,65]. Values of Ky less than 1 indicate that the decrease in PDW and GY is generally not significant due to the decline in water supply, whereas values more than 1 imply that the decrease in both variables is significant because of the decrease in water supply [66]. In this study, in the first and second seasons, the values of Ky based on PDW were 0.61 and 0.56, respectively, and those based on GY were 0.70 and 0.67, respectively (Figure 7). These values are comparatively lower than those reported by Doorenbos and Kassam [41] for spring wheat (1.15). This finding indicates that the appropriate combinations of PPs and mulching practices can be employed at the study site as an effective management strategy for growing wheat under limited water applications. This speculation is supported by the wide range of Ky values found among the different combinations of PPs and mulching practices under 0.50 ET (Table 3). Table 3 shows that the values of $Ky_{PDW}$ and $Ky_{GY}$ in the two non-mulched treatments (CFNM and RBNM) were significantly higher than those in the three treatments mulched with plastic sheet (CFPM, RBPM, and RFPM) by 56.1–62.9% and 37.8–49.7%, respectively. This indicates that the plastic sheet helped to cushion the negative impacts of the water deficit on the growth and yield of wheat. Additionally, treatments mulched with wheat straw (CFSM and RBSM) helped to cushion the negative impacts of the water deficit on plant growth ($Ky_{PDW} < 1$) but failed to cushion the negative impacts on GY ($Ky_{GY} > 1$) (Table 3). This may be because the effectiveness of wheat straw mulching in reducing evaporation from the soil surface and conserving soil water decreased over time because of its natural decomposition, particularly under arid conditions. These results are consistent with the study of Igbadun and Oiganji [67], who showed that, under low irrigation, the Ky values for mulched treatments of the onion crop were lower than those for non-mulched treatments by 2–13%, and the treatment mulched with plastic sheet helped to cushion the impact of the water deficit on plant yield more than the treatments mulched with rice straw.

The linear relationship of PDW and GY with ET (Figure 8A,B) also implies that the combination of PP and mulching practice has a significant impact on the growth and GY of wheat through indirect influence on ET. As mentioned above, the combination of PP and mulching practice has the ability to reduce evaporation and effectively conserve soil water, thus improving ET. This is consistent with Mak-Mensah et al. [30], who reported that the use of plastic film mulching led to a reduction in ET by an average of 0.5–12.8%. Wen et al. [68] also reported that mulching practices has more impact on soil ET than irrigation scheduling. Therefore, the most important factor explaining the variation in PDW (67% and 61%) and GY (78% and 83%), in the first and second seasons, respectively, under the combination of PPs with mulching practices, was ET in this study (Figure 8A,B). Huang et al. [69] also reported that the most important factor explaining the variation in the GY of wheat was ET

under different irrigation regimes; approximately 66% of the variation in GY was explained by ET alone. El-Hendawy et al. [3] also reported that 64–71% of the variation in wheat GY could be attributed to the variation in ET under different irrigation rates and plant densities, where these treatments had a significant influence on ET by restraining both soil evaporation and canopy transpiration.

*4.4. Crop Water-Production Function*

Improving WUE is the core goal of irrigation studies. In general, the highest WUE can be achieved under a given treatment mainly by reducing ET or increasing GY. Therefore, in most cases, the GY is displayed a significant and linear relationship with WUE. In this study, we observed a nonsignificant relationship between GY and WUE, and the polynomial model was chosen as the best model to describe this relationship when data obtained under two irrigation rates were combined together (Figure 8D). However, when this relationship split according to each irrigation rate, GY displayed a significant and linear relationship ($R^2$ = 0.92 and 0.98 under 1.00 and 0.50 ET, respectively) with WUE (data not shown). These results indicated that WUE varied substantially and significantly among the different PPs, and this variation was dependent on the irrigation rate. Under 1.00 ET, the treatments mulched with wheat straw (CFSM and RBSM) potentially improved several physiochemical and biological proprieties of soil [53–55,57], thus leading to higher GY compared with the other PP treatments and thereby greater WUE. However, under 0.50 ET, the treatments mulched with plastic film (CFPM, RBPM, and RFPM) made better use of the limited water supply by reducing ET and conserving more soil moisture [30,49,51,63,70,71], thus improving water availability to wheat plants compared with other PP treatments, which led to higher GY and thereby greater WUE. Therefore, high WUE under 1.00 ET depends on the ability of the PP to achieve high GY, whereas high WUE under 0.50 ET depends on the ability of the PP to reduce ET and conserve more soil moisture. This may explain why WUE was not necessarily associated with GY when the data of both irrigation rates were combined together (Figure 8D), while the relationship between GY and WUE was strong for each irrigation rate separately (data not shown).

To confirm this finding, the relationships of different wheat parameters with the SWC measured prior to the next irrigation at different soil depths under 1.00 ET and 0.50 ET were tested separately. We observed a quadratic and nonsignificant relationship between SWC at different soil depths and all parameters under 1.00 ET, with a few exceptions (Figure 9). By contrast, linear and strong correlations were detected between the SWC at different soil depths and most wheat parameters under 0.50 ET ($R^2$ range = 0.68 to 0.99; Figure 10). Additionally, the total SWC in the 0–60 cm soil layer measured before irrigation exhibited a strong relationship with all parameters (Table 4). These findings further confirm that enhancement of growth, production, and WUE of wheat under limited irrigation water supply directly depends on the ability of agronomic practices to conserve more soil moisture, and the changes in SWC could result in corresponding linear changes in the performance of wheat under different agronomic practices, such as the combination of PP and mulching practice. Therefore, agronomic practices that effectively conserve soil water are essential for optimizing the performance of wheat and for obtaining high GY and WUE under limited water supply.

## 5. Conclusions and Outlook

Water shortage limits wheat production in arid and semiarid regions, mainly because of scarce rainfall and limited availability of other water resources. Thus, it is important to develop feasible and effective strategies that can sustain wheat production and efficient irrigation water use in these regions. In this study, we investigated the impacts of combining modified PPs with different mulching practices on SWC, growth, GY, and WUE of spring wheat. This is the first time that such a study has been conducted in Saudi Arabia, a typical arid country. We found that the different PPs and mulching practices had significant effects on SWC in the top 60 cm soil layer, and the PPs mulched with plastic sheet (RFPM, CFPM,

and RBPM) or wheat straw (CFSM and RBSM) significantly increased the SWC by 0.4–34.8% and 0.6–20.2%, respectively, when compared with the two non-mulched PPs (CFNM and RBNM) under 0.50 ET. The impacts of tested treatments on growth, yield components, GY, and WUE varied with the irrigation rate. The PPs mulched with wheat straw (CFSM and RBSM) were more effective in improving these measurements under 1.00 ET, while PPs mulched with plastic film (RFPM, CFPM, and RBPM) were helpful in improving these measurements under 0.50 ET. These findings were confirmed by the Ky data; the PDW- and GY-based Ky values in PPs mulched with plastic film were less than one under 0.50 ET. Production functions also revealed weak and strong relationships under 1.00 and 0.50 ET, respectively, between the measured parameters and the SWC in the 0–60 cm soil layer. Therefore, combining modified PPs with plastic film mulching can serve as a feasible and effective strategy for sustaining wheat production in arid and semiarid irrigation regions.

Even though the PPs mulched with the plastic film were more effective than those mulched with wheat straw in improving the growth, GY, and WUE of wheat under limited irrigation rate in this study, numerous previous individual studies have reported that the residual plastic film may degrade the fertility and quality of soil as well as causing serious problems in pollution of the soil and water resources. Additionally, the extra cost of applying plastic film including plastic film machinery and manpower may also decrease the net economic returns of this technique. Therefore, the economic and environmental aspects of implementing plastic film mulching have to be further considered, especially under arid conditions. It is important to develop the materials of plastic sheeting to minimize its negative impacts on soil and the environment, while maximizing its benefits; thus, further studies are urgently needed to achieve this goal.

**Author Contributions:** Conceptualization, S.E.-H., Y.R., B.A., M.A.M. and E.T.; methodology, S.E.-H., B.A., N.A.-S., N.M., M.A. and Y.R.; software, S.E.-H., Y.R., B.A., N.M. and E.T.; validation, S.E.-H., Y.R., M.A., N.A.-S., M.A.M. and E.T.; formal analysis, S.E.-H., B.A., N.M., Y.R., M.A., M.A.M. and E.T.; investigation, S.E.-H., Y.R., B.A., M.A.M. and E.T.; resources, S.E.-H., Y.R., E.T. and M.A.M.; data curation, S.E.-H., B.A., N.M., M.A., N.A.-S., E.T., M.A.M. and Y.R.; writing—original draft preparation, S.E.-H.; writing—review and editing, S.E.-H.; visualization, S.E.-H., M.A.M., N.A.-S., Y.R. and E.T.; supervision, S.E.-H.; project administration, S.E.-H.; funding acquisition, S.E-H. All authors have read and agreed to the published version of the manuscript.

**Funding:** This project was funded by the National Plan for Science, Technology, and Innovation (MAARIFAH), King Abdul-Aziz City for Science and Technology, Kingdom of Saudi Arabia, Award Number (12-AGR2901-02).

**Institutional Review Board Statement:** Not applicable.

**Informed Consent Statement:** Not applicable.

**Data Availability Statement:** All data are presented within the article.

**Acknowledgments:** The authors extend their appreciation to the National Plan for Science, Technology, and innovation (MAARIFAH) at the King Abdul-Aziz City for Science and Technology, Saudi Arabia for funding this work through Award Project No. (12-AGR2901-02).

**Conflicts of Interest:** The authors declare no conflict of interest.

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
