# Peer review of "Combining Planting Patterns with Mulching Bolsters the Soil Water Content, Growth, Yield, and Water Use Efficiency of Spring Wheat under Limited Water Supply in Arid Regions"

_agronomy, doi:10.3390/agronomy12061298_

Round 1
Reviewer 1 Report
The manuscript reports on combining planting patterns with mulching bolsters the soil water content, growth, yield and water use efficiency of spring wheat under limited water supply. Although a good job has been done, some corrections are still needed. Some of my substantial concerns are listed below:
- The objectives of the study seem not clear. The authors need to revise the abstract section.
-Introduction: Please add more literature review. -Examples of studies conducted in the same area with different methods can be added.
-The entire analyses of this study are relatively simple and more information, and statistical analysis is necessary to support this study. Results are not done with detailed information.
- Authors are advised to enhance the discussion on results.
-The authors should polish the results and highlight the most important conclusions.
- There are several grammatical mistakes, therefore, it is recommended that the authors work with a science editor who is proficient in the Native English language to improve the organization and grammatical structure of the manuscript.
Author Response
Reviewer #1
The manuscript reports on combining planting patterns with mulching bolsters the soil water content, growth, yield and water use efficiency of spring wheat under limited water supply. Although a good job has been done, some corrections are still needed. Some of my substantial concerns are listed below:
Response: We greatly appreciate your critical observations as well as your constructive and helpful comments. We hope that we could address your questions/comments by the explanations and revisions made in the manuscript. We believe that the manuscript is substantially improved after making the suggested revisions.
- The objectives of the study seem not clear. The authors need to revise the abstract section.
Response: Thank you very much for your comment. The objectives of this study have been modified and the abstract section has been revised.
-Introduction: Please add more literature review. -Examples of studies conducted in the same area with different methods can be added.
Response: Thank you very much for your suggestion. Several articles published in 2020-2022 have been cited in the introduction section. The following references are example for these citations. The introduction section has been revised according your comments.
Sun, D.; Li, H.; Wang, E.; He, W.; Hao, W.; Yan, C.; Li, Y.; Xurong Mei, X.; Zhang, Y.; Sun, Z.; Jia, Z.; Zhou, H.; Fan, T.; Zhang, X.; Liu, Q.; Wang, F.; Zhang, C.; Shen, J.; Wang, Q.; Zhang, F. An overview of the use of plastic-film mulching in China to increase crop yield and water-use efficiency. National Sci. Rev.2020, 7, 1523–1526.
Cheng, D., Wang, Z., Yang, L., Zhang, L., Zhang, Q. Combined effects of mulching and crop density on soil evaporation, temperature, and water use efficiency of winter wheat. Exp. Agric. 2021, 57(3), 163-174.
Noor, M.A.; Nawaz, M.M.; Ma, W.; Zhao, M. Wheat straw mulch improves summer maize productivity and soil properties. Italian J. Agron. 2021, 16,1623.
Mak-Mensah, E.; Obour, P.B.; Essel, E.; Wang, Q.; Ahiakpa, JK. Influence of plastic film mulch with biochar application on crop yield, evapotranspiration, and water use efficiency in northern China: A meta-analysis. PeerJ, 2021, 9:e10967.
Zhang, R.; Lei, T.; Wang, Y.; Xu, J.; Zhang, P.; Han, Y.; Hu, C.; Yang, X.; Sadras, V. Zhang, S. Responses of yield and water use efficiency to the interaction between water supply and plastic film mulch in winter wheat-summer fallow system. Agric. Water Manage. 2022, 266, 107545.
Gao, H.; Yan, C.; Liu, Q.; Li, Z.; Yang, X.; Qi, R. Exploring optimal soil mulching to enhance yield and water use efficiency in maize cropping in China: A meta-analysis. Agric. Water Manage. 2019, 255, 105741
Xie, J.; Wang, L.; Li, L.; Anwar, S.; Luo, Z.; Zechariah, E.; Kwami Fudjoe, S. Yield, Economic Benefit, Soil Water Balance, and Water Use Efficiency of Intercropped Maize/Potato in Responses to Mulching Practices on the Semiarid Loess Plateau. Agriculture 2021, 11, 1100.
-The entire analyses of this study are relatively simple and more information, and statistical analysis is necessary to support this study. Results are not done with detailed information.
Response: Thank you very much for your comment. To test the different objectives of this study, several statistical analyses have been applied such as analysis of variance (ANOVA), appropriate for a split-plot design, Duncan’s multiple range test to test the significance of the difference between treatment means, and Linear and quadratic regression analyses to examine the relationship between the different measured parameters and SWC at different soil depths. Furthermore, the appropriate combination between PPs and mulching with plastic film and wheat straw to achieve maximum GY and WUE simultaneously for irrigated spring wheat under arid conditions has been tested through yield response factor (Ky), crop water-yield production function (CWPF), and WUE–yield relationships.
- Authors are advised to enhance the discussion on results.
Response: Thank you very much for your comment. The discussion section has been improved and the results of this study have been discussed under the following subtitles:
- Effects of PPs and Mulching Practices on SWC
- The Impacts of PPs and Mulching Practices on Growth, Yield, and WUE Depending on Irrigation Rate
- Identify the Optimal Coupling Combinations between PPs and Mulching Practices for Enhancing Growth, GY, and WUE under Limitted Water Applications
- Crop-Water Production Function
-The authors should polish the results and highlight the most important conclusions.
Response: Thank you very much for your comment. The conclusion section has highlighted the most important results of this study.
- There are several grammatical mistakes, therefore, it is recommended that the authors work with a science editor who is proficient in the Native English language to improve the organization and grammatical structure of the manuscript.
Response: Thank you very much for your suggestion. The manuscript has been carefully reviewed by an experienced editor whose first language is English and who specializes in editing papers written by scientists whose native language is not English (certifies below).

Reviewer 2 Report
This study has a certain guiding significance for the production, I think it is very interesting, the following opinions are for reference.
- Important literatures published in 2020-2022 should be included in the introduction to define state of art work in the field. The introduction should be improved to better frame the scope of the work.
Get to the point – hypothesis, context, approach, rationale
- In terms of content presentation and flow of written text.
- A thorough scientific editing is mandatory for the manuscript.
Author Response
Reviewer #2
This study has a certain guiding significance for the production, I think it is very interesting, the following opinions are for reference.
Response: We greatly appreciate your critical observations as well as your constructive and helpful comments. We hope that we could address your questions/comments by the explanations and revisions made in the manuscript. We believe that the manuscript is substantially improved after making the suggested revisions.
- Important literatures published in 2020-2022 should be included in the introduction to define state of art work in the field. The introduction should be improved to better frame the scope of the work.
Response: Thank you very much for your suggestion. Several articles published in 2020-2022 have been cited in the introduction section. The following references are example for these citations. The introduction section has been revised according your comments.
Sun, D.; Li, H.; Wang, E.; He, W.; Hao, W.; Yan, C.; Li, Y.; Xurong Mei, X.; Zhang, Y.; Sun, Z.; Jia, Z.; Zhou, H.; Fan, T.; Zhang, X.; Liu, Q.; Wang, F.; Zhang, C.; Shen, J.; Wang, Q.; Zhang, F. An overview of the use of plastic-film mulching in China to increase crop yield and water-use efficiency. National Sci. Rev.2020, 7, 1523–1526.
Cheng, D., Wang, Z., Yang, L., Zhang, L., Zhang, Q. Combined effects of mulching and crop density on soil evaporation, temperature, and water use efficiency of winter wheat. Exp. Agric. 2021, 57(3), 163-174.
Noor, M.A.; Nawaz, M.M.; Ma, W.; Zhao, M. Wheat straw mulch improves summer maize productivity and soil properties. Italian J. Agron. 2021, 16,1623.
Mak-Mensah, E.; Obour, P.B.; Essel, E.; Wang, Q.; Ahiakpa, JK. Influence of plastic film mulch with biochar application on crop yield, evapotranspiration, and water use efficiency in northern China: A meta-analysis. PeerJ, 2021, 9:e10967.
Zhang, R.; Lei, T.; Wang, Y.; Xu, J.; Zhang, P.; Han, Y.; Hu, C.; Yang, X.; Sadras, V. Zhang, S. Responses of yield and water use efficiency to the interaction between water supply and plastic film mulch in winter wheat-summer fallow system. Agric. Water Manage. 2022, 266, 107545.
Gao, H.; Yan, C.; Liu, Q.; Li, Z.; Yang, X.; Qi, R. Exploring optimal soil mulching to enhance yield and water use efficiency in maize cropping in China: A meta-analysis. Agric. Water Manage. 2019, 255, 105741
Xie, J.; Wang, L.; Li, L.; Anwar, S.; Luo, Z.; Zechariah, E.; Kwami Fudjoe, S. Yield, Economic Benefit, Soil Water Balance, and Water Use Efficiency of Intercropped Maize/Potato in Responses to Mulching Practices on the Semiarid Loess Plateau. Agriculture 2021, 11, 1100.
- A thorough scientific editing is mandatory for the manuscript.
Response: Thank you very much for your suggestion. The manuscript has been carefully reviewed by an experienced editor whose first language is English and who specializes in editing papers written by scientists whose native language is not English (certifies below).

Reviewer 3 Report
- The manuscript is in depth study about Combining Planting Patterns with Mulching Bolsters in Arid Regions. The authors did a set of physiological studies in order to shed light on the Water Use Efficiency of Spring Wheat under Limited Water Supply. This is a important field of study and I do think that this manuscript has important and empirical data.
Regarding the amount of data presented, some changes is needed in order to achieved a better organization of the manuscript. For example the figure 2 is not well formatted, the numbers are from the left to the right, not a regular way to present images.
About the results I do recommend authors to remove the massive description of the figures (pencentages) and include in a supplementary files.
Also, I suggest the authors discuss the environmental damage caused by the massive use of plastic in the agriculture. If in on hand a benefits for cover the soil prevent the water loss and improve the amount of water in the above ground, on the other hand this method could be novice to the enviroment. I would like the authors discussed this topic.
Author Response
Reviewer #3
The manuscript is in depth study about Combining Planting Patterns with Mulching Bolsters in Arid Regions. The authors did a set of physiological studies in order to shed light on the Water Use Efficiency of Spring Wheat under Limited Water Supply. This is an important field of study and I do think that this manuscript has important and empirical data.
Response: We greatly appreciate your critical observations as well as your constructive and helpful comments. We hope that we could address your questions/comments by the explanations and revisions made in the manuscript. We believe that the manuscript is substantially improved after making the suggested revisions.
- Regarding the amount of data presented, some changes are needed in order to achieve a better organization of the manuscript. For example the figure 2 is not well formatted; the numbers are from the left to the right, not a regular way to present images.
Response: Thank you very much for your suggestion. Figure 2 has been reformatted.
- About the results I do recommend authors to remove the massive description of the figures (percentages) and include in a supplementary files.
Response: Thank you very much for your suggestion. Because the impact of the combination of planting patterns (PPs) and mulching with plastic film and wheat straw on soil water content (SWC) at different soil depths and at three-time points (before irrigation, 7 days after irrigation, and 14 days after irrigation) is very important, it is important also to present and discuss all results of this point in the body of MS.
- Also, I suggest the authors discuss the environmental damage caused by the massive use of plastic in the agriculture. If in on hand a benefits for cover the soil prevent the water loss and improve the amount of water in the above ground, on the other hand this method could be novice to the environment. I would like the authors discussed this topic.
Response: Thank you very much for this important comment. Further information about the negative impacts of residual plastic film on soil and environment has been provided.

Round 2
Reviewer 3 Report
The authors really improved the manuscript. I wish success with this publication.